# Feature Attribution for Label-Specific Feature Selection in Multi-Label Learning

## Abstract

Multi-label dimensionality reduction is an important but difficult problem. Feature selection approach searches a feature subset for multiple labels, which preserves label correlations well; however, not all features are necessary for specific labels. Label-specific approach constructs diverse feature spaces for different labels, paying more concentration on label characteristics but facing the complexity pressure arising from the number of labels. In this paper, a feature attribution based label-specific feature selection method is proposed, striking a balance between efficiency and accuracy. Feature attribution quantifies the contribution of every feature to the prediction, which commonly can be achieved with the gradient of deep learning output with respect to input. For multi-label learning, a simple shallow neural network is sufficient to construct a multi-input multi-output mapping, where the gradient of each label with respect to each input feature can be used for label-specific feature ranking and selection. Compared to state-of-the-art multi-label feature selection methods, the proposed method achieves comparable or superior performance while requiring less than 10% of their runtime in most cases.

## 1 Introduction

Multi-label learning, where each instance can be associated with multiple labels simultaneously, arises in a wide range of real-world applications such as text categorization, image annotation, and item recommendation Li et al. (2025). However, the high-dimensional feature spaces in such tasks often lead to increased training costs and overfitting risks, highlighting the importance of effective feature selection to improve learning efficiency and generalization performance.

In recent years, numerous multi-label feature selection algorithms have been proposed. They employ perspectives such as feature redundancy, information theory Yang et al. (2025); Qian et al. (2021), causal learning Wu et al. (2020), graph-based modeling Zhang et al. (2024); Fan et al. (2025a), etc., selecting a competent feature subset for all labels to consider label correlations. Nonetheless, features that are effective for a distinct label are not necessarily useful for other labels, that is, multi-label feature selection fails to maximize the compression of dimensionality. Conversely, label-specific methods Zhang & Wu (2014); Zhan & Zhang (2017) (formally known as: multi-label learning with label-specific features) reconstruct feature spaces for different labels, and for each label, a binary classifier can be trained to handle it.

In comparison, the feature selection approach ultimately produces a multi-label learning dataset suitable for multi-label learners, while the label-specific approach results in multiple single-label datasets, and the number of newly generated features can theoretically be smaller. However, the label-specific approach induces separate analysis and dimensionality reduction for each label, leading to higher computational complexity.

In this paper, a fast label-specific feature selection method FALS (namely, Feature Attribution based Label-specific Selection) is proposed, which employs feature attribution to simultaneously identify the optimal feature subsets for all labels. Feature attribution Simonyan et al. (2013); Ancona et al. (2017) is a technique from explainable AI, typically taking the gradient of output with respect to input to evaluate the model's sensitivity to different features: If a feature has a large gradient, it is indicated as a high contribution feature as its slight change is able to affect the prediction.

In FALS, gradient of a multi-layer perceptron (MLP) is used to evaluate feature scores. As a multi-input multi-output (MIMO) system, the MLP can simultaneously establish the mapping from all features to all labels, and based on the trained parameters, several gradient computations are able to obtain feature scores and rankings for each label. It is worth noting that our task is to identify the most sensitive features rather than to train a high-accuracy multi-label classifier; therefore, a shallow neural network with weak performance can accomplish this goal quickly.

To summarize, the contributions of this paper include:

- A weak network (2-layer MLP) suffices for feature attribution in multi-label learning and enables effective feature selection.
- Both theoretical analysis and experimental results validate that label-specific feature selection is superior to conventional feature selection in multi-label learning.
- FALS achieves performance comparable to using the full feature set while retaining only a small fraction (10%) of the features, matching or surpassing state-of-the-art dimensionality reduction methods at a speed far surpassing them (runtime less than 10% in most cases and 1% in many cases).

## 2 PRELIMINARIES

Formally, multi-label learning refers to constructing a MIMO mapping $\boldsymbol{f} : \mathcal{X} \to \mathcal{Y}$ from feature space $\mathcal{X} \in \mathbb{R}^D$ to label space $\mathcal{Y} \in \{0,1\}^C$, where $0,1$ represent negative and positive classes, respectively. We further specialize $X_d, 1 \leq d \leq D$ denoting the $d$-th feature, $Y_c, 1 \leq c \leq C$ denoting the $c$-th label, and $f_c(\boldsymbol{x}) : \mathbb{R}^D \to \{0,1\}$ denoting $\boldsymbol{f}(\boldsymbol{x})$'s predictor for $Y_c$ given an instance $(\boldsymbol{x}, \boldsymbol{y}) \in (\mathcal{X}, \mathcal{Y})$. In empirical model training, we take $\{(\boldsymbol{x}^n, \boldsymbol{y}^n), 1 \leq n \leq N\}$ representing the multi-label dataset. Besides, $k$ is the number of selected features and $H$ is the number of hidden neurons.

In the following text, we use lowercase letters to denote scalars, and their boldface for vectors. Uppercase letters are used for matrices or some important scalar quantities.

## 3 THE PROPOSED METHOD

In FALS, the gradient of a multi-label neural network is employed to enable feature attribution. Specifically for instance $(\boldsymbol{x}, \boldsymbol{y})$, the gradient of label $Y_c$ with respect to feature $X_d$ is taken to measure feature score $S_{cd}$. The technique framework of FALS is depicted as Fig. 1, where gradient values are represented by the heights and colors of rectangular blocks.

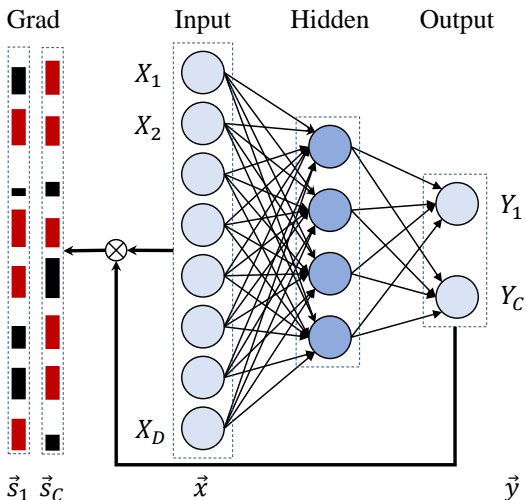

Figure 1: FALS: Generating label-specific feature scores via a 2-layer perceptron.

## 3.1 TWO-LAYER PERCEPTRON

A simple two-layer perceptron (TLP) is designed to perform feature attribution. Since our task is to evaluate the impact of input features on the output, it is unnecessary to build a highly accurate neural network, instead, a shallow model facilitates fast feature selection. The rationale behind TLP design will be explained in detail.

In TLP, the numbers of neurons in input and output layer are clearly $D$ and $C$, respectively. For hidden layer, the number of neurons needs to be sufficiently large to extract latent patterns, but excessively high dimensionality may lead to overfitting. An empirical approach is to set the number of hidden layer neurons between that of the input and output layers. In FALS, the dimension of hidden layer is set as $H = \sqrt{DC}$ to ensure proportional scaling of information from input to output.

At the end of hidden layer, ReLU is applied to introduce nonlinearity, due to its advantages of simplicity and efficiency. And at the output layer, a Sigmoid function is applied to compress the output values into the range $[0, 1]$, representing the predicted probability of each label.

To summarize, the TLP model can be formulated as:

$$\boldsymbol{f}(\boldsymbol{x}) = \sigma\left(\phi(\boldsymbol{x}W_1 + \boldsymbol{b}_1)W_2 + \boldsymbol{b}_2\right) \tag{1}$$

where $\boldsymbol{x} \in \mathbb{R}^D$ is an input vector, $W_1 \in \mathbb{R}^{D \times H}, W_2 \in \mathbb{R}^{H \times C}$ are the coefficient matrices in two layers and $\boldsymbol{b}_1 \in \mathbb{R}^H, \boldsymbol{b}_2 \in \mathbb{R}^C$ are the corresponding bias vectors, and $\sigma(\cdot)$ denotes probability mapping function:

$$\sigma(x) = \text{Sigmoid}(x) = \frac{1}{1 + e^{-x}} \tag{2}$$

and $\phi(\cdot)$ denotes nonlinear activate function:

$$\phi(x) = \text{ReLU}(x) = max\{0, x\} \tag{3}$$

## 3.2 LABEL-SPECIFIC FEATURE SCORES

According to the chain rule of differentiation, the gradient matrix of TLP's output with respect to its input is $G \in \mathbb{R}^{C \times D}$:

$$G(\boldsymbol{x}) = \frac{\partial \boldsymbol{f}(\boldsymbol{x})}{\partial \boldsymbol{x}} = \text{diag}\left(\boldsymbol{f}(\boldsymbol{x}) \odot (1 - \boldsymbol{f}(\boldsymbol{x}))\right) W_2^T \text{diag}\left(\mathbb{I}(\boldsymbol{x}W_1 + \boldsymbol{b}_1)\right) W_1^T \tag{4}$$

where $\mathbb{I}(\cdot)$ is an indicator function representing the gradient of ReLU:

$$\mathbb{I}(x) = \begin{cases} 1, & \text{if } x > 0 \\ 0, & \text{if } x \leq 0 \end{cases} \tag{5}$$

Specifically, $c$-th row of $G$ is the gradient of $Y_c$ with respect to $\boldsymbol{x}$:

$$\boldsymbol{g}_c = \frac{\partial f_c(\boldsymbol{x})}{\partial \boldsymbol{x}} = f_c(\boldsymbol{x})(1 - \boldsymbol{f}_c(\boldsymbol{x}))W_2^T \text{diag}\left(\mathbb{I}(\boldsymbol{x}W_1 + \boldsymbol{b}_1)\right) W_1^T \tag{6}$$

More specifically, $G_{cd}$, the gradient of $Y_c$ to $X_d$, implies the effect of the perturbation in $d$-th feature on the $c$-th label, can be positive or negative. To directly describe the influence level, we use squared gradients to form a feature attribution matrix $A \in (\mathbb{R}^+)^{C \times D}$:

$$A(\boldsymbol{x}) = G(\boldsymbol{x}) \odot G(\boldsymbol{x}) \tag{7}$$

where $\odot$ denotes Hadamard product.

The feature attribution matrix in Eq. (7) admits a direct connection to Fisher information. Let $F^c(\boldsymbol{x}) \in \mathbb{R}^{D \times D}$ denote the Fisher information matrix of $f_c(\boldsymbol{x})$ with respect to $\boldsymbol{x}$:

$$F^c(\boldsymbol{x}) = \mathbb{E}_{Y_c|X=\boldsymbol{x}}\left[(\nabla_{\boldsymbol{x}} \log f_c(Y_c \mid X = \boldsymbol{x}))^2\right] \tag{8}$$

which measures the information that $\boldsymbol{x}$'s $D$ features can provide for predicting $Y_C$. Accordingly, Fisher information matrix of $\boldsymbol{f}(\boldsymbol{x})$ with respect to the features of $\boldsymbol{x}$ is:

$$F(\boldsymbol{x}) = \mathbb{E}_{\mathcal{Y}|X=\boldsymbol{x}}\left[(\nabla_{\boldsymbol{x}} \log \boldsymbol{f}(\mathcal{Y} \mid X = \boldsymbol{x}))^2\right] = \sum_{c=1}^{C} F^c(\boldsymbol{x}) \tag{9}$$

The relation between $F^c$ and $\boldsymbol{a}_c$ provides a statistical interpretation to evaluate and identify the information content of individual features:

$$\mathrm{diag}(F^c(\boldsymbol{x})) = \frac{\boldsymbol{a}_c}{f_c(\boldsymbol{x})(1 - f_c(\boldsymbol{x}))} \tag{10}$$

Furthermore, a basic inequality below demonstrates that label-specific selection can provide more information than joint multi-label feature selection:

$$\sum_{c=1}^{C} \max_k \omega_c \boldsymbol{a}_c = \sum_{c=1}^{C} \max_k \mathrm{diag}(F^c(\boldsymbol{x})) \geq \max_k \mathrm{diag}(F(\boldsymbol{x})) = \max_k \sum_{c=1}^{C} \omega_c \boldsymbol{a}_c \tag{11}$$

where $\omega_c = \frac{1}{f_c(\boldsymbol{x})(1 - f_c(\boldsymbol{x}))}$ and $\max_k\{\cdot\}$ denotes selecting the most large $k$ values from the vector.

The detailed relationship between Fisher information and feature attribution matrices can be seen in Appendix C.

In practice, the real gradient and feature attribution matrices are unknown. We approximately take the empirical mean of instance-wise gradients over the training set to get the final feature score matrix $S \in (\mathbb{R}^+)^{C \times D}$:

$$S = \frac{1}{N} \sum_{n=1}^{N} A(\boldsymbol{x}^n) \tag{12}$$

where $S_{cd}$ is the effectiveness score of $X_d$ for $Y_c$.

For convenience, we define the $c$-th row of $A$ and $S$ as $\boldsymbol{a}_c$ and $\boldsymbol{s}_c$, specific to label $Y_c$. Eventually for each label, the features with top-$k$ scores are selected as the label-specific features. The whole workflow is summarized as Alg. 1.

---

**Algorithm 1** Framework of FALS.

---

**Input**: Training data $\{\boldsymbol{x}^n, \boldsymbol{y}^n\}, 1 \leq n \leq N$, parameter $k$.
**Output**: Selected features $\{F_c\}, 1 \leq c \leq C$.
 1: Training TLP to get the model Eq. (1).
 2: **for** $1 \leq n \leq N$ **do**
 3:     Compute gradient matrix $G(\boldsymbol{x}^n)$ via Eq. (4).
 4:     Compute attribution matrix $A(\boldsymbol{x}^n)$ via Eq. (7).
 5: **end for**
 6: Compute feature score matrix $S$ via Eq. (12).
 7: **for** $1 \leq c \leq C$ **do**
 8:     Get top-$k$ features for $Y_c$, $F_c = \mathrm{argmax}(\boldsymbol{s}_c, k)$.
 9: **end for**
10: **Return**: $\{F_c\}, 1 \leq c \leq C$.

---

### 3.3 COMPLEXITY ANALYSIS

TLP implements a fast multi-label learner and feature attribution evaluator, whose time and space complexities are $O(TN(DH+HC))$ and $O(H(D+C))$, where $T$ is the number of epochs (iteration rounds). Existing multi-label dimensionality reduction approaches can be divided into several types.

Firstly among regular feature selection methods, information-based, Relief-based and tree-based algorithms measure feature competence in a certain order. Take decision tree as an example, it performs feature selection a time complexity of $O(CDN \log N)$ and a space complexity of $O(N)$. While linear models like LASSO select features with high weights, exhibiting time complexity $O(TNDC)$ and space complexity $O(ND)$.

Secondly, the family of LIFT, the popular label-specific approach, executes $k$-means clustering for each label to generate label-specific features. Assume that $T$ iterations are required for $k$-means, the time complexity of clustering is $O(TkDN)$ and LIFT has a time complexity of $O(CTkDN)$ with time consumed outside of clustering is negligible, while the space complexity of clustering and LIFT are $O(ND + kD)$ and $O(ND + kD + NC)$, respectively.

Thirdly, multi-label feature selection (MLFS) methods Qian et al. (2021); Zhang et al. (2023a; 2024); Dai et al. (2024); Fan et al. (2025b) generally discriminate the pairwise separability between instances to find the significant features, and for each instance pair, $D$ features for $C$ labels should be analyzed. Therefore, the time and space complexities are $O(T(N^2DC))$ and $O(N^2(D + C))$.

To summarize, the time and space complexities of common dimensionality reduction methods are listed as Tab. 1.

Table 1: Time and space complexities of common methods, where * indicates time-efficient or memory-safe.

|  | Time | Space |
|---|---|---|
| Tree | $O(CDN \log N)$* | $O(N)$* |
| LASSO | $O(TNDC)$* | $O(ND)$* |
| LIFT | $O(TkNDC)$ | $O(ND + NC + D^2)$ |
| MLFS | $O(T(N^2DC))$ | $O(N^2(D + C))$ |
| FALS | $O(TN(D^{\frac{3}{2}}C^{\frac{1}{2}} + D^{\frac{1}{2}}C^{\frac{3}{2}}))$* | $O(D^{\frac{3}{2}}C^{\frac{1}{2}} + D^{\frac{1}{2}}C^{\frac{3}{2}})$* |

According to the comparison, tree and LASSO methods that perform feature selection separately for label individuals yields limited time and space complexity, however, they neglect the characteristics of multi-label learning. LIFT family utilizes clustering to generate label-specific features, and MLFS methods consider the intrinsic relationships among instances, labels, and features, making them less efficient when applied to large-scale data. Furthermore, some MLFS methods analyze pairwise label correlation, which may increase the corresponding $C$ in Tab. 1 to $C^2$.

In contrast, FALS is friendly to time and memory. First, typical optimization methods (MLFS and clustering in LIFT) require 100–1000 iterations, whereas FALS only needs $T = 10$. Second, multi-label learning commonly collects fewer features than instances $D < N$ and fewer labels than features, $C < D$, so that the memory usage of FALS actually lower than that of feature and label matrices in size of $ND + NC$. That is, as long as a computer is capable of loading the data, it is sufficient to run FALS; however, while LIFT and MLFS may be reported out of memory during execution. Besides, FALS can be easily executed on GPU with mini-batch training and very low memory consumption, which is benefits large-scale datasets. The comparison of runtime will be reported in Experiments.

## 4 EXPERIMENTS

### 4.1 EXPERIMENTAL SETUP

We take 10 public datasets to evaluate the performance of FALS, as listed in Tab. 2.

Table 2: Benchmark datasets.

| DataId | Dataset | Domain | N | D | C |
|---|---|---|---|---|---|
| #1 | 20NG | Text/News | 19300 | 1006 | 20 |
| #2 | Corel5k | Image | 5000 | 499 | 374 |
| #3 | HumanGO | Biology | 3106 | 9844 | 14 |
| #4 | Ohsumed | Text/Medical | 13930 | 1002 | 23 |
| #5 | Tmc2007_500 | Text | 28600 | 500 | 22 |
| #6 | Yelp | Text/Recommend | 10810 | 671 | 5 |
| #7 | Bibtex | Text/Bibliography | 7395 | 1836 | 159 |
| #8 | Bookmarks | Text/Recommend | 87860 | 2150 | 208 |
| #9 | Delicious | Text/Recommend | 16110 | 500 | 983 |
| #10 | Imdb | Text/Movie | 120900 | 1001 | 28 |

FALS is compared with 1 baseline and 9 feature reduction algorithms, categorized as 4 types:

- 1 baseline: using full rather than selected features.

- 3 single-label selection methods: random forest (RF), LASSO, and GEFS Liu et al. (2024).

- 2 label-specific methods: LIFT Zhang & Wu (2014) and LACE Zhan & Zhang (2017).

- 4 multi-label feature selection methods: FSRD Qian et al. (2021), GLFS Zhang et al. (2023a), LRDG Zhang et al. (2024) and FLFS Zhang et al. (2025).

All experiments are conducted on a laptop equipped with an Intel i7-13650HX CPU with 16GB RAM and an NVIDIA 4060 Laptop GPU with 8GB VRAM.

We take 4 widely-used metrics in the experiments, including Hamming Loss ↓, One Error ↓, Ranking Loss ↓, Average Precision ↑, where the first one evaluates the error rates on individual labels, and the others are ranking-based, evaluating label correlation Madjarov et al. (2012). The detailed description on these metrics are provided in Appendix D.1.

Table 3: Comparison results.

| Metric | DataId | BRDT | RF | LASSO | GEFS | LIFT | LACE | FSRD | GLFS | LRDG | FLFS | FALS |
|---|---|---|---|---|---|---|---|---|---|---|---|---|
| HL ↓ | #1 | 0.0742 | 0.0791 | 0.0782 | 0.0720 | 0.0858 | 0.0763 | 0.0918 | 0.0767 | 0.0752 | 0.0888 | **0.0719** |
| | #2 | 0.0149 | 0.0139 | 0.0109 | **0.0103** | 0.0125 | 0.0123 | 0.0156 | 0.0105 | 0.0106 | 0.0113 | 0.0104 |
| | #3 | **0.0483** | 0.0486 | 0.0501 | 0.0606 | 0.0624 | 0.0643 | 0.0935 | 0.0500 | 0.0828 | 0.0849 | 0.0495 |
| | #4 | 0.0538 | 0.0569 | 0.0562 | 0.0619 | 0.0567 | **0.0490** | 0.0663 | 0.0576 | 0.0607 | 0.0696 | 0.0533 |
| | #5 | **0.0583** | 0.0627 | 0.0661 | 0.0795 | 0.0586 | 0.0605 | 0.0770 | 0.0756 | 0.0830 | 0.0901 | 0.0710 |
| | #6 | **0.1746** | 0.1907 | 0.1835 | 0.2130 | 0.2209 | 0.2230 | 0.2268 | 0.1908 | 0.1998 | 0.2578 | 0.1896 |
| | #7 | 0.0198 | 0.0189 | 0.0210 | 0.0225 | **0.0162** | 0.0165 | 0.0275 | 0.0217 | 0.0295 | 0.0278 | 0.0207 |
| | #8 | 0.0182 | 0.0176 | 0.0172 | 0.0180 | **0.0144** | 0.0179 | 0.0320 | 0.0170 | 0.0181 | 0.0185 | 0.0177 |
| | #9 | 0.0335 | 0.0288 | 0.0237 | **0.0209** | 0.0259 | 0.0248 | 0.0297 | 0.0210 | 0.0226 | 0.0231 | 0.0213 |
| | #10 | 0.0587 | 0.0689 | 0.0607 | 0.0522 | 0.0676 | **0.0465** | 0.0677 | 0.0567 | 0.0467 | 0.0560 | 0.0488 |
| | Avg | 0.0554 | 0.0586 | 0.0568 | 0.0611 | 0.0621 | 0.0591 | 0.0728 | 0.0578 | 0.0629 | 0.0728 | **0.0554** |
| OE ↓ | #1 | 0.3659 | 0.3756 | 0.3650 | 0.3805 | 0.4389 | 0.4416 | 0.3900 | **0.3416** | 0.3470 | 0.4127 | 0.3704 |
| | #2 | 0.8537 | 0.8507 | 0.7518 | 0.8040 | 0.8267 | 0.7980 | 0.8531 | 0.7458 | 0.7920 | 0.7974 | **0.7416** |
| | #3 | 0.2896 | 0.2830 | 0.2944 | 0.2877 | 0.4644 | 0.4815 | 0.5097 | 0.3077 | 0.6743 | 0.5821 | **0.2688** |
| | #4 | 0.7248 | 0.7265 | **0.7097** | 0.7819 | 0.7984 | 0.7943 | 0.8140 | 0.7109 | 0.7515 | 0.8300 | 0.7175 |
| | #5 | 0.2569 | 0.2704 | 0.2737 | 0.3179 | 0.2700 | 0.2759 | 0.3327 | 0.2846 | 0.3487 | 0.3802 | **0.2501** |
| | #6 | 0.2998 | 0.3367 | 0.2931 | 0.2777 | 0.3823 | **0.1969** | 0.4146 | 0.2977 | 0.2614 | 0.3723 | 0.2330 |
| | #7 | 0.6328 | 0.6178 | 0.6280 | 0.7445 | 0.7200 | 0.6911 | 0.8024 | 0.6239 | 0.8150 | 0.7865 | **0.6170** |
| | #8 | 0.7102 | 0.7072 | 0.6945 | 0.7594 | 0.7300 | 0.8271 | 0.8732 | 0.7061 | 0.7412 | 0.7584 | **0.6928** |
| | #9 | 0.6580 | 0.6471 | 0.5877 | 0.5532 | 0.6309 | 0.6188 | 0.6240 | **0.4862** | 0.5259 | 0.5929 | 0.5214 |
| | #10 | 0.8472 | 0.8337 | 0.7715 | 0.8265 | 0.8607 | 0.8946 | 0.8769 | 0.7663 | 0.7388 | 0.8006 | **0.7380** |
| | Avg | 0.5639 | 0.5649 | 0.5369 | 0.5733 | 0.6122 | 0.6020 | 0.6491 | 0.5271 | 0.5996 | 0.6313 | **0.5151** |
| RL ↓ | #1 | 0.2988 | 0.3023 | 0.2770 | 0.2396 | 0.3485 | 0.3354 | 0.3224 | 0.2455 | 0.2430 | 0.2728 | **0.2281** |
| | #2 | 0.3384 | 0.2835 | 0.2066 | 0.2170 | 0.3411 | 0.2867 | 0.2502 | **0.1807** | 0.1919 | 0.1932 | 0.2243 |
| | #3 | 0.1529 | 0.1495 | 0.1376 | **0.1152** | 0.2513 | 0.3409 | 0.2184 | 0.1665 | 0.1928 | 0.2074 | 0.1253 |
| | #4 | 0.4614 | 0.4658 | 0.4561 | 0.4700 | 0.5195 | 0.4431 | 0.5119 | 0.4199 | **0.4162** | 0.4831 | 0.4432 |
| | #5 | 0.1223 | 0.1295 | 0.1018 | 0.1214 | 0.1344 | 0.1244 | 0.1322 | 0.1239 | 0.1195 | 0.1395 | **0.0789** |
| | #6 | 0.2583 | 0.2794 | 0.2454 | 0.2320 | 0.3249 | 0.3266 | 0.3283 | 0.2424 | 0.2058 | 0.2903 | **0.1702** |
| | #7 | 0.3130 | 0.3127 | **0.3012** | 0.3243 | 0.4076 | 0.4698 | 0.3842 | 0.3034 | 0.3682 | 0.3702 | 0.3095 |
| | #8 | 0.3366 | 0.3350 | 0.3166 | 0.3319 | 0.3656 | 0.4077 | 0.3921 | **0.3018** | 0.3289 | 0.3316 | 0.3183 |
| | #9 | 0.3097 | 0.2889 | 0.2391 | 0.1942 | 0.3478 | 0.3375 | 0.2784 | **0.1768** | 0.2019 | 0.2105 | 0.2066 |
| | #10 | 0.5335 | 0.5185 | 0.3900 | 0.3631 | 0.5460 | 0.5669 | 0.5521 | 0.3800 | **0.3127** | 0.4095 | 0.3530 |
| | Avg | 0.3125 | 0.3065 | 0.2671 | 0.2609 | 0.3587 | 0.3639 | 0.3370 | 0.2541 | 0.2581 | 0.2908 | **0.2457** |
| AP ↑ | #1 | 0.5710 | 0.5625 | 0.5858 | 0.5996 | 0.5034 | 0.6192 | 0.5413 | 0.6112 | 0.6070 | 0.5610 | **0.6193** |
| | #2 | 0.1067 | 0.1544 | 0.2088 | 0.1961 | 0.1117 | 0.1679 | 0.1449 | 0.2009 | 0.1989 | 0.1902 | **0.2144** |
| | #3 | 0.7589 | 0.7621 | 0.7692 | 0.7857 | 0.6107 | 0.6284 | 0.5432 | 0.7409 | 0.5151 | 0.5635 | **0.7859** |
| | #4 | 0.3205 | 0.3172 | 0.3295 | 0.2886 | 0.2540 | **0.4088** | 0.2521 | 0.3496 | 0.3280 | 0.2570 | 0.3334 |
| | #5 | 0.7474 | 0.7352 | 0.7630 | 0.6990 | 0.7341 | 0.7245 | 0.6872 | 0.7264 | 0.6635 | 0.6480 | **0.7720** |
| | #6 | 0.7874 | 0.7661 | 0.7946 | 0.7961 | 0.7343 | **0.8918** | 0.7226 | 0.7913 | 0.8101 | 0.7447 | 0.8363 |
| | #7 | 0.3385 | 0.3381 | 0.3443 | 0.2368 | 0.2099 | 0.2371 | 0.1929 | **0.3486** | 0.1923 | 0.1950 | 0.3465 |
| | #8 | 0.3017 | 0.3052 | **0.3209** | 0.2616 | 0.2758 | 0.2873 | 0.2518 | 0.3101 | 0.2815 | 0.2648 | 0.3196 |
| | #9 | 0.1990 | 0.2181 | 0.2561 | 0.2419 | 0.1865 | 0.2426 | 0.1981 | **0.2867** | 0.2634 | 0.2124 | 0.2790 |
| | #10 | 0.2119 | 0.2254 | 0.3221 | 0.2986 | 0.1993 | 0.3537 | 0.1886 | 0.3199 | **0.3553** | 0.2846 | 0.3536 |
| | Avg | 0.4343 | 0.4384 | 0.4694 | 0.4404 | 0.3820 | 0.4561 | 0.3723 | 0.4686 | 0.4215 | 0.3921 | **0.4860** |

Table 4: Dimensionality reduction time (seconds), where GEFS and FALS* are run in GPU.

| DataId | RF | LASSO | GEFS | LIFT | LACE | FSRD | GLFS | LRDG | FLFS | FALS | FALS* |
|--------|------|-------|-------|-------|--------|-------|--------|-------|-------|------|-------|
| #1 | 79 | 1.1 | 136 | 170 | 1728 | 29444 | 1214 | 290 | 87 | 5.8 | 6.6 |
| #2 | 87.4 | 1.9 | 951 | 246 | 10908 | 1312 | 7.1 | 22 | 14 | 5.4 | 1.6 |
| #3 | 29.3 | 1.3 | 100 | 1175 | 99 | 4898 | 170 | 120 | 85 | 1.7 | 1.6 |
| #4 | 45.4 | 1.1 | 168 | 143 | 737 | 15233 | 537 | 152 | 112 | 4.0 | 4.4 |
| #5 | 58.8 | 1.5 | 297 | 76 | 5645 | 42351 | 1015 | 444 | 380 | 6.7 | 8.1 |
| #6 | 7.2 | 0.9 | 30 | 12 | 142 | 7321 | 169 | 67 | 70 | 2.4 | 3.1 |
| #7 | 16.5 | 6.8 | 836 | 1429 | 4754 | 6526 | 137 | 78 | 59 | 6.3 | 2.7 |
| #8 | 885 | 175 | 12980 | 34872 | 234557 | 11653 | 266042 | 16144 | 15554 | 100 | 30 |
| #9 | 162 | 23 | 8074 | 1968 | 197094 | 13521 | 13378 | 298 | 11928 | 91 | 10 |
| #10 | 185 | 11 | 1681 | 1448 | 68089 | 11384 | 84014 | 12800 | 5679 | 40 | 38 |

## 4.2 COMPARISON RESULTS

### 4.2.1 EVALUATION SCORES

According to the recommendation from KIDS website, we perform stratified sampling to create training and test sets with a 2:1 ratio. The fixed datasets split helps to reproduce results.

In this paper, we set the ratio of selected features as $\frac{k}{D} = 0.1$. The comparison experiments are conducted in 2 steps: (1) Feature selection or generation, executed by FALS and comparison methods with training data. (2) Model verification, executed by BRDT (binary relevance decision tree, i.e., a set of $C$ independent classifiers for $C$ labels) with feature-reduced training and testing data.

The evaluation scores of all methods are reported in Tab. 3, where some conclusions can be drawn to validate the strengths of FALS:

- FALS performs best in average scores of 4 metrics.

- Among 40 cases over 4 metrics and 10 datasets, FALS ranks first in 14 cases, surpassing all comparison methods, including 3 for BRDT, 0 for RF, 3 for LASSO, 3 for GEFS, 2 for LIFT, 5 for LIFTACE,0 for FSRD, 7 for GLFS, 3 for LRDG, 0 for FLFS.

### 4.2.2 RUNTIME

To make a fair runtime comparison, we run FALS in CPU and GPU, respectively. And comparison methods except for GEFS are run in CPU. When code running in GPU, some intermediate results still need to be moved to CPU, so that in some small datasets, CPU get shorter runtime than GPU. According to the runtime comparison results shown in Tab. 4, it can be found that:

- FALS has a similar speed to simple algorithms like RF and LASSO, but when the number of labels is very large, such as in Corel5k ($C = 374$), Bookmarks ($C = 208$) and Delicious ($C = 983$), FALS gets some advantages.

- FALS runs significantly faster than the recent methods including gradient-based (GEFS), label-specific learning, and multi-label feature selection methods.

- LIFT and LIFTACE, especially the latter, suffer from extremely slow runtime, making them impractical for real-world applications. This inefficiency stems from the $k$-means clustering for each label, and LIFTACE further incorporating clustering ensemble for each label.

- Multi-label feature selection methods may run very slowly under certain extreme conditions due to the incorporation of quadratic complexity terms related to $N, D, C$, which are introduced to capture the correlations of instances ($N$), features ($D$), labels ($C$), and each two of the three.

- The significant runtime advantage of FALS stems from its use of a simple shallow neural network architecture that computes gradients and obtains feature scores for $C$ labels simultaneously. Besides, GPU can help accelerating big data.

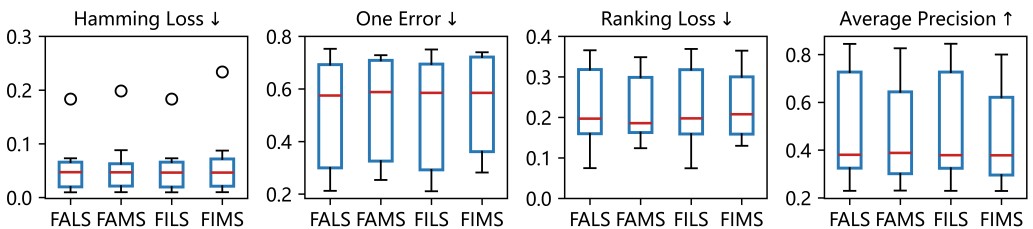

Figure 2: Performance comparison across various feature selection patterns.

### 4.2.3 THE PATTERNS OF FEATURE SELECTION

FALS takes a label-specific pattern to trigger multi-label feature selection. Factually, other patterns of gradient-based feature attribution can be employed. Here, we compare following algorithms:

- FALS (Feature Attribution for Label-specific Selection): Select a feature subset for each label, i.e., taking $\boldsymbol{a}_c$ to evaluate features for $Y_c$.
- FAMS (Feature Attribution for Multi-label Selection): Select a feature subset for all labels, i.e., taking $\sum_{c=1}^{C} \boldsymbol{a}_c$ to evaluate features for each label.
- FILS (Fisher Information for Label-specific Selection): Select a feature subset for each label with Fisher information matrix, i.e., taking $\frac{\boldsymbol{a}_c}{f_c(\boldsymbol{x})(1-f_c(\boldsymbol{x}))}$ to evaluate features for $Y_c$.
- FIMS (Fisher Information for Multi-label Selection): Select a feature subset for all labels with Fisher information matrix,, i.e., taking $\sum_{c=1}^{C} \frac{\boldsymbol{a}_c}{f_c(\boldsymbol{x})(1-f_c(\boldsymbol{x}))}$ to evaluate features for each label.

where the elements of Fisher information-based feature selection can be found in Appendix (Theorem 1 and Theorem 2).

The 4 patterns are compared in Fig. 2. According to the comparison results, following phenomena are in line with our expectations:

- FALS≈FILS and FAMS≈FIMS: For feature selection, Fisher information can be viewed as a variant of feature attribution, so the difference between them is not significant.
- FALS≻FAMS and FILS≻FIMS: For multi-label feature selection, label-specific pattern pay more attention to the characteristics of individual labels.

### 4.3 IN-DEPTH ANALYSIS OF POTENTIAL RISKS IN FALS

FALS employs the gradients of a shallow neural network (two-layer perceptron, TLP) to estimate feature contributions in multi-label learning, which may introduce several potential risks. First, from the perspective of model, TLP is unstable under limited training epochs and may suffer from gradient saturation. Second, from the perspective of task, multi-label learning imposes special requirements in label correlation and extreme minority class, which seems beyond the capacity of a simple network. To dispel these doubts, we examine the gradients, selected features and performance of FALS, its variants, and other feature attribution approaches.

Specifically, the comparable variants of FALS (number of epoch, $L = 10$) include:

- epoch20 / epoch50 / epoch100: FALS with $L = 20, 50, 100$.
- L2-Norm: FALS incorporating L2-norm regularization.
- Dropout: FALS applying dropout.
- FALSb: FALS taking binary classifiers, replacing 1 $c$-label TLP with $c$ 1-label binary TLPs, where each binary TLP possessing $D, \sqrt{D}, 1$ neurons in input, hidden and output layers.

and other representative feature attribution approaches (they pay attention to alleviate or eliminate gradient saturation) include:

- IG (Integrated Gradients) Sundararajan et al. (2017): integrating gradients along a path from a baseline to the input. We set baseline as $x' = 0$ and number of steps as 50.
- DeepLIFT (Deep Learning Important FeaTures) Shrikumar et al. (2017): assigning contributions by comparing activations to a reference baseline.
- LRP (Layer-wise Relevance Propagation) Bach et al. (2015): a backward redistribution technique that propagates prediction relevance through the network layers.

Technical details of these 3 methods are provided in Eq. (15), (19) and (21) in Appendix B.4. And they are implemented using the corresponding functions in the Captum library with Python.

The performance comparison results are reported as the chart boxes in Fig. 3. As for the relation between selected feature subsets, we take 20NG dataset as an example, and illustrate the comparison between FALS and aforementioned methods as the violin plots in Fig. 4, including the Jaccard similarity between two feature sets and the Spearman's rank correlation of two feature confidence vectors. The calculation of these two indices will be supplemented in Appendix D.2. For each dataset, every comparable algorithm is conducted 10 times and average performance is reported.

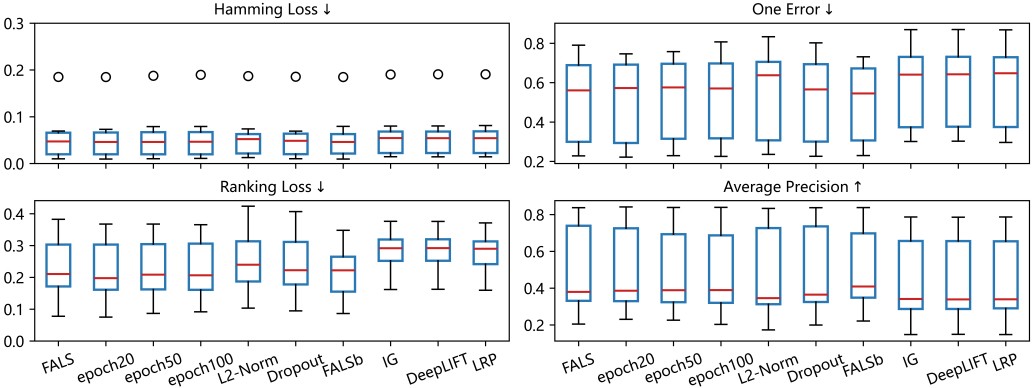

Figure 3: Performance comparison across different FALS settings and feature attribution patterns.

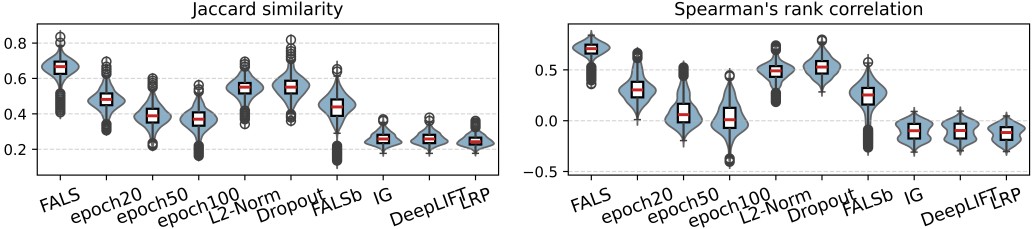

Figure 4: Relations between feature subsets selected by different methods.

### 4.3.1 MODEL ISSUE: TRAINING STABILITY

Let us focus on the first 6 boxes (or violins) of each subfigure in Fig. 3 and Fig. 4.

In Fig. 4, 1st violin of each subfigure illustrates the similarities of features selected by 10 FALS with different random initialization, and other violins are those of similarities between FALS and comparable method. Apparently, 1st violin has the highest values, demonstrating the stability of FALS. Then, 2nd to 4th violins show that with the increasing training epochs, the selected features changes. Conversely, in 5th and 6th violins, the anti-overfitting mechanisms (L2-Norm and Dropout) select more correlated features.

Nonetheless, according to Fig. 3, FALS has a similar accurateness comparing to the others, and in some cases, it gains a slight advantage. The comparison results indicate that FALS properly fits data, that is, no further training is needed, nor is it necessary to avoid overfitting.

### 4.3.2 MODEL ISSUE: GRADIENT SATURATION

Let us focus on the last 3 boxes (or violins) of each subfigure in Fig. 3 and Fig. 4.

According to Fig. 4, FALS selects different features compared other feature attribution approaches. However, the performance comparison in Fig. 3 illustrates FALS's superiority, which can be attributed to the intrinsic compatibility between vanilla gradients and shallow perceptron.

TLP with limited depth exhibits mild nonlinearity, causing the model to behave almost linearly in the local neighborhood of each instance. Under such conditions, the vanilla gradient provides a direct and noise-free estimate of the model's local sensitivity to each feature. In contrast, attribution approaches such as IG, DeepLIFT, and LRP rely on baseline, path integration, or layer-wise redistribution, all of which introduce additional assumptions that not align with the characteristics of high-dimensional tabular inputs and shallow architectures. Factually, these feature attribution approaches mainly work on complex data like images Qi et al. (2020); Patil & Bhat (2024); Binder et al. (2016).

### 4.3.3 TASK ISSUE: LABEL CORRELATION

Label correlation is generally believed to improve the accuracy or efficiency of multi-label models. At first glance, FALS never considers label correlation as it triggers feature selection label by label. However, FALS actually jointly learns the MIMO mapping from $D$ features to $C$ labels, allowing inter-label dependencies to be implicitly captured. To verify this assumption, we compare FALS with FALSb, a variant that independently considers $C$ labels, in which a separate binary TLP is trained for each label and corresponding gradients are taken to select features for this label. According to the 7-th box (violin) of each subfigure in Fig. 3 and Fig. 4, FALS yields noticeably different feature selections compared to FALSb, indicating that label correlation is considered within FALS. To sum up, FALS strikes a balance between label difference and label correlation.

### 4.3.4 TASK ISSUE: CLASS IMBALANCE

We compiled statistics on positive classes and positive gradients for each dataset, and the bar chart in Fig. 5 shows the comparison. On the one hand, multi-label datasets indeed suffer from a severe imbalance, with a low proportion of positive instances. On the other hand, FALS yields a high proportion of positive gradients, indicating that sufficient features beneficial for predicting positive labels are selected. That is, FALS has payed enough attention to minority class (positive label). Therefore, we never perform any further class-balanced design on it.

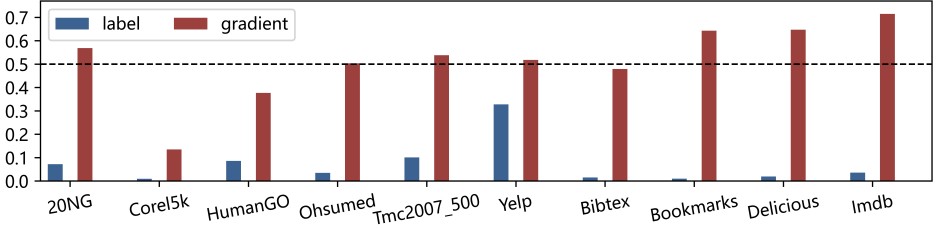

Figure 5: Imbalance ratios of positive labels and gradients.

## 5 CONCLUSIONS

This paper presents FALS, a novel dimensionality reduction method for multi-label learning, which performs label-specific feature selection by leveraging the gradient-based feature attribution technique. FALS offers two major advantages: strong predictive performance and high computational efficiency. It achieves comparable or superior results to SOTA methods while requiring only 10%, 1%, or even less of their runtime, which are demonstrated via both theoretical analysis and extensive experiments. Besides, the lightweight design of FALS not only contributes to its high efficiency but also facilitates model interpretability.

## ETHICS STATEMENT

This work does not present any ethical concerns. It does not involve human subjects, sensitive data, or applications with foreseeable societal harm. All research activities comply with the ICLR Code of Ethics.

## REPRODUCIBILITY STATEMENT

We have taken measures to ensure the reproducibility of our work. The source code has been provided in the Supplementary Material, and clear pseudocode descriptions are included in the main text. These resources together enable independent verification of our experimental results.

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

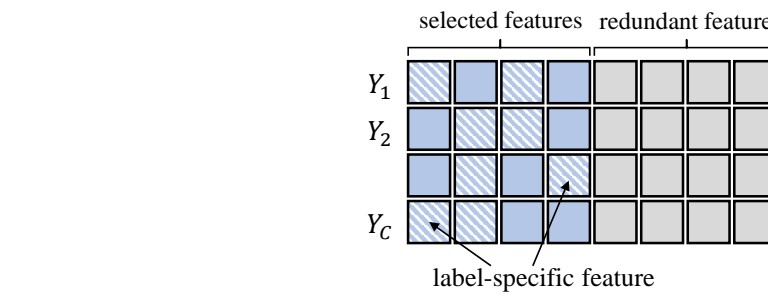

Figure 6: The different between multi-label feature selection and label-specific selection.

## A  THE USE OF LARGE LANGUAGE MODELS (LLMS)

In this work, Large Language Models (LLMs) are employed primarily as writing assistants. Their role is limited to language polishing, such as improving fluency, refining grammar, and checking spelling errors. Importantly, no part of the research design, methodology, data analysis, or result interpretation relies on LLMs. The use of LLMs is restricted to enhancing the readability and presentation quality of the manuscript.

## B  RELATED WORK

### B.1  MULTI-LABEL LEARNING AND FEATURE SELECTION

The earliest multi-label algorithm can be traced back to Binary Relevance (BR), which essentially decomposes a multi-label problem into $C$ independent binary classification tasks. Clearly, it is linear and simple, but fails to consider multi-label structure.

Feature selection is one of the mainstream approaches to dimensionality reduction. Multi-label feature selection, in particular, requires evaluating feature importance while meeting the needs of $C$ labels, especially by leveraging label correlation information. Among existing algorithms, constructing and optimizing multi-objective functions has become a common paradigm, aiming to jointly consider feature discriminability, redundancy, and relevance to labels. For example, some methods focus on label relationship modeling, employing techniques such as graph structures, gain analysis, or manifold learning to capture label dependencies Jian et al. (2016); Zhang et al. (2022); Wang et al. (2018); Li et al. (2022). Other methods emphasize latent representation learning, mapping features and labels into a shared space to capture hidden structures Jian et al. (2016); Hu et al. (2020); Zhang et al. (2024). Some approaches further explore redundancy control mechanisms by applying diverse distance metrics and regularization strategies to enhance feature selection quality Zhang et al. (2022); Fan et al. (2022; 2024). Additionally, certain methods introduce novel ideas such as label-enhanced supervision Fan et al. (2025b), fuzzy theory Qian et al. (2021); Dai et al. (2024); Yang et al. (2025), causal inference Wu et al. (2020), thereby expanding the research frontier of multi-label feature selection. Overall, these methods collectively propel the field from heuristic strategies toward structured modeling and theoretical grounding.

### B.2  MULTI-LABEL LEARNING WITH LABEL-SPECIFIC FEATURES

A common and unavoidable issue in multi-label feature selection is the insufficient reduction on feature dimension. Suppose an algorithm selects $k$ features from the original $D$ ones, the selected features are wanted to cover all labels. However, from a dimensionality reduction perspective, the optimal approach is to select the most valuable $k_i$ features for distinct label $Y_i$. Such difference is depicted in Fig. 6.

LIFT Zhang & Wu (2014) was the first algorithm to generate label-specific features for multi-label learning. Its idea is to cluster the positive and negative instances of each label separately and use the distances from all instances to the cluster centers as new features. In other words, it performs clustering analysis on the instance distribution under each label and the instances within a cluster are

represented with the center instance. Subsequent studies have continuously improved this clustering-based approach by precisely defining the number of clusters Zhang et al. (2014), proposing more stable clustering algorithms Guo et al. (2019); Zhang et al. (2015), and leveraging label correlations to enhance clustering robustness Zhan & Zhang (2017); Guan et al. (2021).

Some algorithms select specific features for each label and merge them into a unified feature subset Zhang et al. (2023b); Yang et al. (2023); Han et al. (2025), so in essence, they are still viewed as multi-label feature selection rather than label-specific approach. Furthermore, some algorithms group labels and perform label-specific feature selection for each label group Zhang et al. (2021); Weng et al. (2023); Zhang et al. (2023a).

### B.3 FEATURE ATTRIBUTION AND FEATURE SELECTION

Feature attribution algorithms aim to explain the predictions of deep learning models for specific inputs by assessing the contribution of each input feature to the final output. Classical methods such as Saliency Maps Simonyan et al. (2013) directly compute the output's gradient with respect to the input, while Integrated Gradients (IG) Sundararajan et al. (2017) estimate gradient along a path from baseline to input, mitigating the vanishing gradient issue. Numerous IG variants further optimize the path, starting point, or integration method to enhance the stability and reliability of the explanation Xu et al. (2020); Miglani et al. (2020); Kapishnikov et al. (2021); Sturmfels et al. (2020). Modified backpropagation methods prune gradients during backward passes to highlight most critical activation paths Zeiler & Fergus (2014); Springenberg et al. (2014); Kim et al. (2019). Some ensemble methods aggregates gradients from multiple sources or perturbed samples to improve attribution robustness Srinivas & Fleuret (2019); Smilkov et al. (2017).

These methods differ in focus: gradient-based approaches are generally well-suited for structured inputs like images or tabular data, while path-integrated methods emphasize stability and precision. More complex methods offer stronger attribution capabilities and are often used in image-based visualization tasks to generate heatmaps. Factually, it is unnecessary to handle tabular data with complex deep learning models Grinsztajn et al. (2022).

Beyond model explanation, several recent studies have explored leveraging feature attribution or gradient-based sensitivity analysis for feature selection. Typical approaches directly rely on the sensitivity of the loss or prediction to input features, such as computing averaged input–gradient magnitudes across samples Peng et al. (2024). Others combine gradient information with pruning or gating strategies to adaptively control which input features remain active during training Zimmer (2024); Wydmański & Śmieja (2025). These methods illustrate the usefulness of gradient signals as feature importance scores, yet they usually depend on a single model snapshot. In contrast, GradEnFS Liu et al. (2024) (abbreviated as GEFS in main text) enhances robustness by aggregating gradient information from an ensemble of sparse neural networks across multiple training stages, yielding more stable feature rankings. In contrast, this paper applies feature attribution to multi-label feature selection, emphasizing its adaptability to multi-label learning and, in particular, its ability to support label-specific feature selection.

### B.4 ATTRIBUTION METHODS BEYOND RAW GRADIENTS

This subsection introduces 3 feature attribution approaches with weak or no dependence on gradients, including IG (Integrated Gradients) Sundararajan et al. (2017), DeepLIFT (Deep Learning Important FeaTures) Shrikumar et al. (2017) and LRP (Layer-wise Relevance Propagation) Bach et al. (2015).

IG is a more refined gradient-based feature attribution method. It calculates the cumulative change of gradients as the model moves from the baseline input to the real input, thus obtaining more robust and less noisy feature importance scores. Denote baseline as $\boldsymbol{x}'$, the score of $d$-th feature can be formulated as:

$$\text{IG}_d(\boldsymbol{x}) = (\boldsymbol{x} - \boldsymbol{x}') \int_0^1 \frac{\partial f(\boldsymbol{x}' + \alpha(\boldsymbol{x} - \boldsymbol{x}'))}{\partial \boldsymbol{x}_d} d\alpha \tag{13}$$

In practice, it is approximated using $m$ integration steps:

$$\text{IG}_d(\boldsymbol{x}) \approx (\boldsymbol{x} - \boldsymbol{x}')\frac{1}{m}\sum_{k=1}^{m}\frac{\partial f(\boldsymbol{x}' + \frac{k}{m}(\boldsymbol{x} - \boldsymbol{x}'))}{\partial \boldsymbol{x}_d} \tag{14}$$

In this paper, we set baseline with all zero, $\boldsymbol{x}' = \boldsymbol{0} \in \mathbb{R}^D$, and the feature scores of $D$ features for $C$ labels are:

$$\text{IG}(\boldsymbol{x}) \approx \boldsymbol{x} \odot \frac{1}{m}\sum_{k=1}^{m}\frac{\partial \boldsymbol{f}(\frac{k}{m}\boldsymbol{x})}{\partial \boldsymbol{x}} = \boldsymbol{x} \odot \frac{1}{m}\sum_{k=1}^{m}G(\frac{k}{m}\boldsymbol{x}) \tag{15}$$

where $\boldsymbol{g}_c(\cdot)$ is from Eq. (4).

DeepLIFT is a feature attribution method based on hierarchical backpropagation, measuring the contribution of input features to the output. It also introduces a reference baseline, but instead of using the gradients, it assigns contribution scores by comparing the model's activation differences between input and baseline. Therefore, DeepLIFT avoids the gradient saturation of Sigmoid and the dead zone of ReLU. The goal of DeepLIFT is to make the sum of all features' contributions equals to the change in the output relative to the baseline, $\sum_{d=1}^{D}C_{\Delta x_d \to \Delta y} = \Delta y$. Suppose an example network with 3 layers $x, h, y$ and the activation for $h$ is $a$, the scores of input are chained together, and the $d$-th score is:

$$S_d = S_{\Delta x_d \to \Delta y} = \sum_{i=1}^{|h|}S_{\Delta x_d \to \Delta h_i} \cdot S_{\Delta h_i \to \Delta a_i} \cdot S_{\Delta h_i \to \Delta y} \tag{16}$$

where the propagation of scores is different in the fully connection layer and the activation layer:

$$S_{\Delta x_d \to \Delta h_i} = W_{di} \tag{17}$$

$$S_{\Delta h_i \to \Delta a_i} = \frac{\Delta a_i}{\Delta h_i} \tag{18}$$

In this paper, we take $\boldsymbol{x}' = \boldsymbol{0}$ as baseline, and the feature scores of $D$ features for $C$ labels are:

$$S(\boldsymbol{x}) = \boldsymbol{x} \odot W_1 diag(\frac{\Delta a}{\Delta h})W_2\frac{\Delta f(\boldsymbol{x})}{W_2^T \Delta a} \tag{19}$$

LRP propagates the relevance of the model output from the output layer back to the input layer, layer by layer, ensuring that the relevance of each layer is conserved, that is, the total relevance of all nodes in each layer is equal to the total relevance of the previous layer. For example, for $y = xW + b$, the contribution of input layer is:

$$R_i(x) = \frac{z_i}{\sum_k z_k + \epsilon}R(y) \tag{20}$$

where $z_i = x_i W_i^+$, $W^+$ denotes that only positive values are employed, and $\epsilon > 0$ is a smooth parameter.

In this paper, the contribution scores of $D$ features for $C$ labels can be formulated as:

$$R(\boldsymbol{x}) = f(\boldsymbol{x}) \odot \boldsymbol{x} \odot \frac{W_1^+}{\boldsymbol{s}^1}\frac{\boldsymbol{h} \odot W_2^+}{\boldsymbol{s}^2} \tag{21}$$

where $\boldsymbol{s}_k^1 = \sum_d x_d W_{1,dk}^+, 1 \le k \le H$ and $\boldsymbol{s}_c^2 = \sum_k h_k W_{2,kc}^+, 1 \le c \le C$.

## C  THEORETICAL APPENDIX

In this section, we first take Bernoulli distribution to formulate binary classification and empirical multi-label learning in Lemma 1 and Lemma 2, and introduce some necessary properties. Then, Definition 1 introduces Fisher information matrix. Theorem 1 discusses the connection between feature attribution matrix and Fisher information matrices for individual labels (which can be viewed as multiple single-labels). Next, Theorem 2 exhibits the connection between feature attribution matrix and multi-label Fisher information matrix. Finally, based on the conclusions of aforementioned theorems, the proposed method is analyzed through the views of Cramér–Rao lower bound and mutual information in Corollary 1 and Corollary 2, respectively.

**Lemma 1** (Binary Classification with Bernoulli Distribution Presentation). *Let $Y \in \{0, 1\}$ denote the label of a binary classification task. Conditioned on the input $X = \boldsymbol{x}$, assume $Y$ follows a Bernoulli distribution with success probability $p(\boldsymbol{x}), 0 \leq p(\boldsymbol{x}) \leq 1$:*

$$(Y \mid X = \boldsymbol{x}) \sim \text{Bernoulli}(p(\boldsymbol{x})) \tag{22}$$

*Then the conditional probability mass function, expectation, and variance are given by:*

$$\Pr(Y = y \mid X = \boldsymbol{x}) = p(\boldsymbol{x})^y (1 - p(\boldsymbol{x}))^{1-y} \tag{23}$$

$$\mathbb{E}[Y \mid X = \boldsymbol{x}] = p(\boldsymbol{x}) \tag{24}$$

$$\text{Var}(Y \mid X = \boldsymbol{x}) = p(\boldsymbol{x})(1 - p(\boldsymbol{x})) \tag{25}$$

**Lemma 2** (Empirical Multi-Label Learning with Bernoulli Distribution Presentation). *Let $\mathcal{Y} = (Y_1, \ldots, Y_C) \in \{0, 1\}^C$ denote the label vector in multi-label learning. Conditioned on the input $X = \boldsymbol{x}$, each component $Y_c, 1 \leq c \leq C$ is assumed to follow a Bernoulli distribution with success probability $p_c(x)$:*

$$(Y_c \mid X = \boldsymbol{x}) \sim \text{Bernoulli}(p_c(\boldsymbol{x})) \tag{26}$$

*In practice, since the true probabilities $\{p_c(\boldsymbol{x})\}$ are unknown, we approximate them with predictive outputs $\boldsymbol{f}(\boldsymbol{x}) \in \mathbb{R}^C$:*

$$\boldsymbol{f}(\boldsymbol{x}) = (p_1(\boldsymbol{x}), \cdots, p_C(\boldsymbol{x})) \tag{27}$$

*Assuming conditional independence across labels, the joint distribution factorizes as:*

$$\Pr(\mathcal{Y} = \boldsymbol{y} \mid X = \boldsymbol{x}; \boldsymbol{f}) = \prod_{c=1}^{C} f_c(\boldsymbol{x})^{y_c} (1 - f_c(\boldsymbol{x}))^{1-y_c} \tag{28}$$

*Moreover, the conditional expectation and variance for each label are:*

$$\mathbb{E}[Y_c \mid X = \boldsymbol{x}; \boldsymbol{f}] = f_c(\boldsymbol{x}) \tag{29}$$

$$\text{Var}(Y_c \mid X = \boldsymbol{x}; \boldsymbol{f}) = f_c(\boldsymbol{x})(1 - f_c(\boldsymbol{x})) \tag{30}$$

**Definition 1** (Fisher Information Matrix). *For a parametric model with likelihood $p(Y \mid X; \theta)$, the Fisher information matrix conditioned on $X = \boldsymbol{x}$ is defined as:*

$$F(x; \theta) = \mathbb{E}_{Y \mid X = \boldsymbol{x}} \left[ (\nabla_\theta \log p(Y \mid X = \boldsymbol{x}; \theta))^2 \right] \tag{31}$$

**Theorem 1** (Fisher Information and Label-Specific Feature Attribution). *Based on Lemma 2 and the multi-label learner $\boldsymbol{f}(\boldsymbol{x})$ defined in Eq. (1), the Fisher information matrix for label $Y_c, 1 \leq c \leq C$ is $F^c \in \mathbb{R}^{D \times D}$:*

$$F^c(\boldsymbol{x}) = \frac{\boldsymbol{g}_c^T \boldsymbol{g}_c}{f_c(1 - f_c)} \tag{32}$$

*Let denote $c$-th row of feature attribution matrix $A$ as $\boldsymbol{a}_c$. The diagonal vector of Fisher information matrix $F^c$ can be viewed as weighted $\boldsymbol{a}_c$, and the weight is related to prediction confidence:*

$$\text{diag}(F^c) = \frac{\boldsymbol{a}_c}{f_c(1 - f_c)} \tag{33}$$

*Proof.* We start from the conditional likelihood of label $Y_c$ under the multi-label predictor $\boldsymbol{f}(\boldsymbol{x})$:

$$p(y_c \mid \boldsymbol{x}; \boldsymbol{f}) = f_c(\boldsymbol{x})^{y_c} (1 - f_c(\boldsymbol{x}))^{1-y_c} \tag{34}$$

Then, $\log p(y_c \mid \boldsymbol{x}; \boldsymbol{f})$ can be formulated as:

$$\log p(y_c \mid \boldsymbol{x}; \boldsymbol{f}) = y_c \log f_c(\boldsymbol{x}) + (1 - y_c) \log(1 - f_c(\boldsymbol{x})) \tag{35}$$

and its derivative with respect to $\boldsymbol{x}$ is:

$$\nabla_{\boldsymbol{x}} \log p(y_c \mid \boldsymbol{x}; \boldsymbol{f}) = \nabla_{\boldsymbol{x}} f_c(\boldsymbol{x}) \left( \frac{y_c}{f_c(\boldsymbol{x})} - \frac{1 - y_c}{1 - f_c(\boldsymbol{x})} \right) \tag{36}$$

$$= \boldsymbol{g}_c(\boldsymbol{x}) \left( \frac{y_c - f_c(\boldsymbol{x})}{f_c(\boldsymbol{x})(1 - f_c(\boldsymbol{x}))} \right) \tag{37}$$

where $\nabla_{\boldsymbol{x}} f_c(\boldsymbol{x}) = \boldsymbol{g}_c(\boldsymbol{x})$ due to Eq. (6).

According to Definition 1, Fisher information matrix for $Y_c$ can be formulated as:

$$F^c(\boldsymbol{x}) = \mathbb{E}\left[(\nabla_{\boldsymbol{x}} \log p(\boldsymbol{y} \mid \boldsymbol{x}; \boldsymbol{f}))^2\right] \tag{38}$$

$$= \frac{\boldsymbol{g}_c^T(\boldsymbol{x})\boldsymbol{g}_c(\boldsymbol{x})}{f_c^2(\boldsymbol{x})(1 - f_c(\boldsymbol{x}))^2} \mathbb{E}\left[(y_c - f_c(\boldsymbol{x}))^2\right] \tag{39}$$

$$= \frac{\boldsymbol{g}_c^T(\boldsymbol{x})\boldsymbol{g}_c(\boldsymbol{x})}{f_c(\boldsymbol{x})(1 - f_c(\boldsymbol{x}))} \tag{40}$$

where let $\ell_c := y_c - f_c(\boldsymbol{x})$, $\mathbb{E}[\ell_c^2] = \text{Var}[\ell_c] - \mathbb{E}^2[\ell_c] = f_c(\boldsymbol{x})(1 - f_c(\boldsymbol{x}))$ due to Eq. (29) and Eq. (30).

According to the feature attribution matrix $A$ defined in Eq. (4), it can be find that its $c$-th row $\boldsymbol{a}_c, 1 \le c \le C$ is:

$$\boldsymbol{a}_c = \boldsymbol{g}_c(\boldsymbol{x}) \odot \boldsymbol{g}_c(\boldsymbol{x}) \tag{41}$$

which proves that $\text{diag}(F^c(\boldsymbol{x})) = \frac{\boldsymbol{a}_c}{f_c(\boldsymbol{x})(1 - f_c(\boldsymbol{x}))}$. $\qquad\square$

**Theorem 2** (Fisher Information and Multi-Label Feature Attribution). *The Fisher information matrix for multi-label learner $\boldsymbol{f}(\boldsymbol{x})$ is:*

$$F(\boldsymbol{x}) = \sum_{c=1}^{C} F^c(\boldsymbol{x}) \tag{42}$$

*And the diagonal vector of $F$ can be viewed as a weighted sum of $\boldsymbol{a}_c, 1 \le c \le C$:*

$$\text{diag}(F(\boldsymbol{x})) = \sum_{c=1}^{C} \frac{\boldsymbol{a}_c}{f_c(\boldsymbol{x})(1 - f_c(\boldsymbol{x}))} \tag{43}$$

*Proof.* Under the assumption that $C$ labels are independent, we can start from the conditional likelihood of the labels under the multi-label predictor $\boldsymbol{f}(\boldsymbol{x})$:

$$p(\boldsymbol{y} \mid \boldsymbol{x}; \boldsymbol{f}) = \prod_{c=1}^{C} f_c(\boldsymbol{x})^{y_c}(1 - f_c(\boldsymbol{x}))^{1-y_c} \tag{44}$$

Then, $\log p(\boldsymbol{y} \mid \boldsymbol{x}; \boldsymbol{f})$ can be formulated as:

$$\log p(\boldsymbol{y} \mid \boldsymbol{x}; \boldsymbol{f}) = \sum_{c=1}^{C} y_c \log f_c(\boldsymbol{x}) + (1 - y_c)\log(1 - f_c(\boldsymbol{x})) \tag{45}$$

and its derivative with respect to $\boldsymbol{x}$ is:

$$\nabla_{\boldsymbol{x}} \log p(\boldsymbol{y} \mid \boldsymbol{x}; \boldsymbol{f}) = \sum_{c=1}^{C} \nabla_{\boldsymbol{x}} f_c(\boldsymbol{x})\left(\frac{y_c}{f_c(\boldsymbol{x})} - \frac{1 - y_c}{1 - f_c(\boldsymbol{x})}\right) \tag{46}$$

$$= \sum_{c=1}^{C} \boldsymbol{g}_c(\boldsymbol{x})\left(\frac{y_c - f_c(\boldsymbol{x})}{f_c(\boldsymbol{x})(1 - f_c(\boldsymbol{x}))}\right) \tag{47}$$

According to Definition 1, Fisher information matrix for multi-label learner can be formulated as:

$$F(\boldsymbol{x}) = \mathbb{E}\left[(\nabla_{\boldsymbol{x}} \log p(\boldsymbol{y} \mid \boldsymbol{x}; \boldsymbol{f}))^2\right] \tag{48}$$

$$= \mathbb{E}\left[\left(\sum_{c=1}^{C} \frac{\boldsymbol{g}_c(\boldsymbol{x})(y_c - f_c(\boldsymbol{x}))}{f_c(\boldsymbol{x})(1 - f_c(\boldsymbol{x}))}\right)^2\right] \tag{49}$$

$$= \mathbb{E}\left[\sum_{1 \le c,d \le C} \frac{\boldsymbol{g}_c^T(\boldsymbol{x})\boldsymbol{g}_d(\boldsymbol{x})(y_c - f_c(\boldsymbol{x}))(y_d - f_d(\boldsymbol{x}))}{f_c(\boldsymbol{x})(1 - f_c(\boldsymbol{x}))f_d(\boldsymbol{x})(1 - f_d(\boldsymbol{x}))}\right] \tag{50}$$

Since label independence, $\mathbb{E}[Y_c Y_d \mid \boldsymbol{x}] = \mathbb{E}[Y_c \mid \boldsymbol{x}]\mathbb{E}[Y_d \mid \boldsymbol{x}] = f_c f_d$, we ask for the expectation of cross-error term $\ell_c \ell_d, \ell_c = y_c - f_c(\boldsymbol{x})$:

$$\mathbb{E}_{\boldsymbol{y}|\boldsymbol{x}}[\ell_c \ell_d] = \begin{cases} \mathbb{E}_{\boldsymbol{y}|\boldsymbol{x}}\left[\ell_c^2\right] = f_c(1 - f_c) & c = d \\ 0 & c \neq d \end{cases} \tag{51}$$

Therefore, the Fisher information matrix can be eventually formulated as $F \in \mathbb{R}^{D \times D}$:

$$F(\boldsymbol{x}) = \mathbb{E}\left[\sum_{c=1}^{C} \frac{\boldsymbol{g}_c^T(\boldsymbol{x})\boldsymbol{g}_c(\boldsymbol{x})(y_c - f_c(\boldsymbol{x}))^2}{f_c^2(\boldsymbol{x})(1 - f_c(\boldsymbol{x}))^2}\right] \tag{52}$$

$$= \sum_{c=1}^{C} \frac{\boldsymbol{g}_c^T(\boldsymbol{x})\boldsymbol{g}_c(\boldsymbol{x})}{f_c(\boldsymbol{x})(1 - f_c(\boldsymbol{x}))} \tag{53}$$

$$= \sum_{c=1}^{C} F^c(\boldsymbol{x}) \tag{54}$$

Next, Eq. (33) of Theorem 1 can complete the proof. $\square$

**Corollary 1** (Cramér–Rao Bound and Multi-label Feature Selection). *For an unbiased estimator $\hat{\boldsymbol{x}}$ of the input features $\boldsymbol{x}$, the covariance is lower-bounded by the inverse of the Fisher information matrix:*

$$\text{Cov}(\hat{\boldsymbol{x}}) \succeq F(\boldsymbol{x})^{-1} \tag{55}$$

*As established in Theorem 1, the input Fisher information is related to feature attribution score matrix $A(\boldsymbol{x})$, which implies that the Cramér–Rao bound for feature $X_d$ under label $Y_c$ is inversely related to attribution score:*

$$\text{Var}(X_d) \geq \frac{1}{F_{dd}} \sim \frac{1}{A_{cd}} \tag{56}$$

*This indicates that features with large attribution scores correspond to high Fisher information and low Cramér–Rao bounds, making them more identifiable and reliably selected.*

**Corollary 2** (Mutual Information and Multi-label Feature Selection). *The mutual information between input $\mathcal{X}$ and output $Y_c$ can be approximated using the input Fisher information:*

$$I(\mathcal{X}; Y_c) \approx \frac{1}{2}\mathbb{E}_{\boldsymbol{x}}\left[\text{Tr}(F^c(\boldsymbol{x})\Sigma_{\boldsymbol{x}})\right] \tag{57}$$

*where $\Sigma_{\boldsymbol{x}}$ is the covariance of the input.*

*With the conclusion of Theorem 1, the mutual information between input $X_d$ and output $Y_c$ can be approximated as:*

$$I(X_d; Y_c) \approx \frac{1}{2}\mathbb{E}_{\boldsymbol{x}}\left[\text{Tr}(F_{dd}^c \Sigma_{\boldsymbol{x}})\right] \sim \mathbb{E}_{\boldsymbol{x}}\left[\text{Tr}(A_{cd}\Sigma_{\boldsymbol{x}})\right] \tag{58}$$

*This shows that selecting features with high attribution scores effectively preserves those that carry the most information about the outputs, providing a theoretical justification for the proposed multi-label feature selection method.*

## D EXPERIMENTAL APPENDIX

### D.1 EVALUATION METRICS DESCRIPTION

Let us denote $Y \in \{0, 1\}^{N \times C}$ as the ground-truth label matrix. For classification-based metrics, we denote $\hat{Y} \in \{0, 1\}^{N \times C}$ as the prediction.

Hamming loss is the most commonly used multi-label classification metric, which considers different labels with equal weights and calculates the prediction error rate for all $N \times C$ label classes:

$$\text{HL} = \frac{1}{N}\frac{1}{C}\sum_{n=1}^{N}\sum_{c=1}^{C}\mathbb{I}(Y_{nc} \neq \hat{Y}_{nc}) \tag{59}$$

For ranking-based metrics, we denote $\hat{Y} \in \mathbb{R}^{N \times C}$ as the prediction.

One Error focuses on the situation where the predicted probability value is the highest in each instance, but it is not actually a positive class:

$$\text{OE} = \frac{1}{N} \sum_{n=1}^{N} \mathbb{I}(Y_{nk} \neq 1), k = argmax_c \hat{Y}_{nc} \tag{60}$$

Ranking Loss considers all positive and negative class label pairs for each instance, and then calculates the proportion of misclassified pairs:

$$\text{RL} = \frac{1}{N} \sum_{n=1}^{N} \frac{|(p, q)|Y_{np} = 0, Y_{nq} = 1, \hat{Y}_{np} < \hat{Y}_{nq}|}{|Y_n = 0||Y_n = 1|} \tag{61}$$

where $|Y_n = 0|, |Y_n = 1|$ are the numbers of positive and negative labels in instances $x_n$, and $\{(p, q)\}$ is the index set of misclassified label pairs.

Average Precision calculates the number of positive labels that rank higher than positive labels and calculates the ratio:

$$\text{AP} = \frac{1}{N} \sum_{n=1}^{N} \frac{1}{|Y_n = 1|} \sum_{k, Y_{nk}=1} \frac{|\{Y_{nc} = 1, \hat{Y}_{nc} > \hat{Y}_{nk}\}|}{rank_{nk}} \tag{62}$$

where $rank_{nk}$ represents the rank of $\hat{Y}_{nk}$ in $\hat{Y}_n$.

## D.2 FEATURE SELECTION SIMILARITY INDEX

To measure the similarity between features selected by two algorithms, two indices are employed.

Jaccard similarity measures the overlap degree of two sets. For each feature to be evaluated, Jaccard similarity never mind its ranking or score, but whether it is selected. For two feature sets, $F, F'$

$$\text{Jac} = \frac{|F \cap F'|}{|F \cup F'|} \tag{63}$$

Spearman's rank correlation measures the consistency degree of two rankings. For feature selection, it considers the evaluation scores of all features. Let $R = \{R_1, \cdots, R_D\}$ represent the rank of each feature in all features, and $R' = \{R'_1, \cdots, R'_D\}$ be another feature rank.

$$\rho = \frac{cov(R, R')}{\sigma_R \sigma_{R'}} = 1 - \frac{6 \sum_{d=1}^{D} (R_d - R'_d)^2}{D(D^2 - 1)} \tag{64}$$

In this paper, only 10% features are selected to take part in final decision, implying the irrationality that take the ranks of all $D$ features into consideration. Therefore, we take $F \cup F'$ to replace the original feature subset, and evaluate Spearman's rank correlation in $D'$ features, $D' = |F \cup F'|$.

$$\rho' = \frac{cov(R, R')}{\sigma_R \sigma_{R'}} = 1 - \frac{6 \sum_{d=1}^{D'} (R_d - R'_d)^2}{D'(D'^2 - 1)} \tag{65}$$

## D.3 ADDITIONAL EXPERIMENTS

### D.3.1 SIGNIFICANCE TEST

We conduct Wilcoxon signed-rank test across all algorithm pairs, considering their comparable performance. The results are visualized in Fig. 7, where circled 9 represents the proposed FALS method, and circles 1 to 8 represent comparison methods. If two algorithms are significantly correlated ($p \geq 0.1$), the 2 corresponding circles are connected by a thick line, and no line for statistically significance ($p < 0.05$), thin line for weak relationship ($0.05 \leq p < 0.1$). The hypothesis testing reveals that:

- For Hamming Loss, the most of algorithms do not exhibit statistically significant differences, implying that feature selection cannot directly reduces model error.

- For ranking-base metrics, the performance differences among algorithms become more statistically significant, which demonstrates the importance of ranking-based metrics for multi-label learning: they possess greater discriminative ability.

- FALS demonstrates statistically significant improvements over most comparison methods.

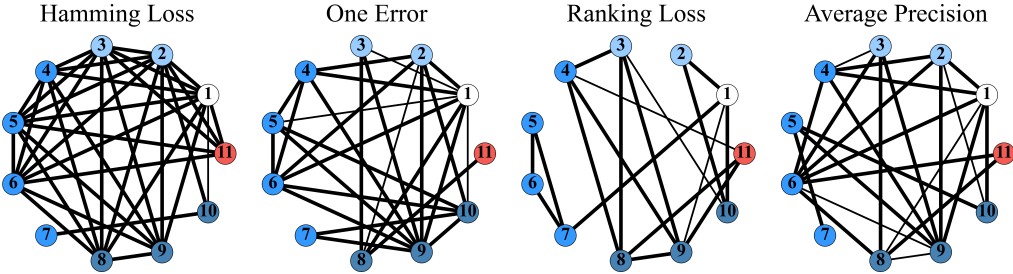

Figure 7: Wilcoxon test results.

### D.3.2 THE NUMBER OF SELECTED FEATURES $k$

The number of selected features directly determines the performance of multi-label learner. We previously set $k = \frac{1}{10}D$; here, we vary it evenly from $k = \frac{1}{10}D$ to $k = \frac{7}{10}D$ to examine its effect on the final results. In Fig. 8, we report the average performance on 10 datasets of FALS and 3 multi-label feature selection methods. The comparison results show that:

- Multi-label learning clearly suffers from feature redundancy. SOTA algorithms including GLFS and FALS obtain good performance with 10% features and model performance decreases with an increasing number of features.

- As the number of features increases, the evaluation scores of different methods tend to be overlapped, because closer and closer selected features lead to similar performance.

- FALS remains the first in terms of the varying numbers of selected features.

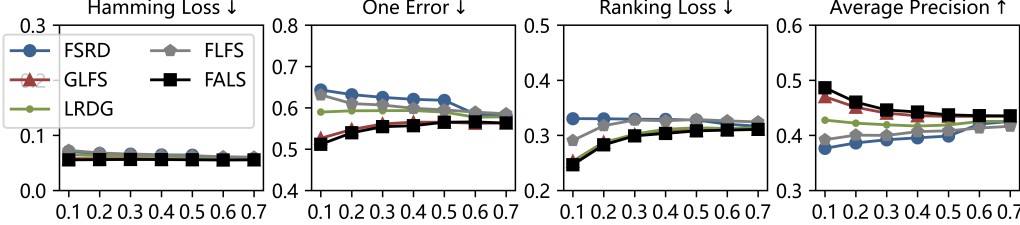

Figure 8: Performance comparison across various feature selection ratios.

### D.3.3 THE ARCHITECTURE OF NEURAL NETWORK

The neural network determines the gradients and feature attribution scores, whose architecture parameters include the number of perceptron layers, the number of hidden neurons and the nonlinear activation function. In this subsection, we test the effect of different parameters on the results. Specifically, activation function are selected to be ReLU or Sigmoid and the layer number is adjusted from 1 to 5, where 1-layer case takes no hidden layers and in other cases, the hidden neurons in different layers are set to meet the principle of proportional scaling from input to output. At the setting of 2-layer and ReLU, the hidden neuron number is adjusted from $H = \{D, C, \sqrt{DC}\}$.

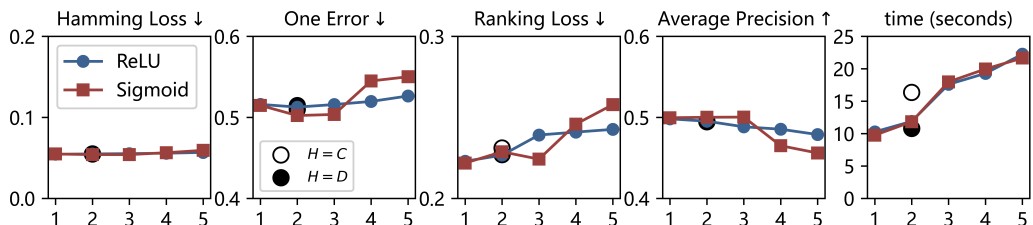

Figure 9: Performance comparison across various perceptron architectures.

The comparison results are shown in Fig. 9. In each sub-figure, the horizontal axis represents the number of neural network layers, and when it is 2, we use black dashed and solid circles to represent $C$ and $D$ hidden neurons, respectively. We can observe that:

- The change in evaluation score is not significant, especially for ReLU, its change never exceed 0.01. This is because as the number of layers increases, Sigmoid encounters the risk of gradient disappearance, while ReLU is more robust. Considering that the proposed method may be used in more complex scenarios, we choose ReLU as the activation function in this paper.

- As the number of layers increases, the time increases linearly. Under comprehensive consideration, we finally choose to set the number of layers to 2.

- In terms of the number of hidden neurons in 2-layer mode, $H = \sqrt{DC}$ performs better than $H = D$ or $H = C$, as an overly aggressive compression mapping may create an information bottleneck, hindering effective feature extraction.

## D.4 SUPPLEMENTARY 10-FOLD CROSS-VALIDATION RESULTS

To further demonstrate the robustness and general applicability of the proposed FALS algorithm, we provide additional results using 10-fold cross-validation. These results confirm that FALS consistently achieves competitive or superior performance compared to baseline feature selection methods across multiple datasets, highlighting its reliability and stability in various multi-label feature selection scenarios.

Considering that many of the baseline multi-label feature selection algorithms require very long training times, we only include comparisons with three recent and representative methods to provide a fair and practical evaluation. The comparison results are reported as following tables, where - indicates that the corresponding algorithm requires more than 100 hours of computation and is therefore not reported.

Table 5: Hamming Loss ↓

| Dataset | GLFS | LRDG | FLFS | FALS |
|---|---|---|---|---|
| 20NG | 0.0691 | 0.0685 | 0.1023 | **0.0675** |
| Corel5k | 0.0140 | 0.0139 | 0.0176 | **0.0095** |
| HumanGO | 0.0455 | 0.0485 | 0.0553 | **0.0447** |
| Ohsumed | 0.0590 | 0.0582 | 0.0656 | **0.0542** |
| Tmc2007_500 | 0.0731 | 0.0716 | **0.0701** | 0.0720 |
| Yelp | 0.1683 | **0.1643** | 0.2073 | 0.1807 |
| Bibtex | **0.0186** | 0.0215 | 0.0286 | 0.0198 |
| Bookmarks | 0.0189 | 0.0211 | 0.0232 | **0.0180** |
| Delicious | 0.0221 | 0.0221 | 0.0291 | **0.0214** |
| Imdb | - | 0.0540 | 0.0606 | **0.0464** |

Table 6: One Error ↓

| Dataset | GLFS | LRDG | FLFS | FALS |
|---|---|---|---|---|
| 20NG | 0.5933 | 0.5931 | 0.7474 | **0.3483** |
| Corel5k | 0.7884 | 0.7918 | 0.8630 | **0.7016** |
| HumanGO | 0.2734 | 0.2727 | 0.4885 | **0.2489** |
| Ohsumed | 0.8538 | 0.8555 | 0.9517 | **0.6589** |
| Tmc2007_500 | 0.2821 | 0.2860 | 0.3057 | **0.2592** |
| Yelp | 0.1664 | **0.1642** | 0.1812 | 0.2111 |
| Bibtex | 0.6022 | 0.6041 | 0.9475 | **0.5871** |
| Bookmarks | 0.9061 | 0.9060 | 0.9528 | **0.6947** |
| Delicious | 0.6235 | 0.6211 | 0.7739 | **0.5261** |
| Imdb | - | 0.9237 | 0.9372 | **0.7287** |

Table 7: Ranking Loss ↓

| Dataset | GLFS | LRDG | FLFS | FALS |
|---|---|---|---|---|
| 20NG | 0.3653 | 0.3658 | 0.5085 | **0.2003** |
| Corel5k | 0.8766 | 0.8818 | 0.9152 | **0.1545** |
| HumanGO | 0.2476 | 0.2461 | 0.4224 | **0.1321** |
| Ohsumed | 0.3904 | 0.3933 | 0.4786 | **0.3709** |
| Tmc2007_500 | 0.3089 | 0.3123 | 0.3403 | **0.0698** |
| Yelp | 0.2839 | 0.2805 | 0.3306 | **0.1517** |
| Bibtex | 0.5923 | 0.5979 | 0.9380 | **0.2936** |
| Bookmarks | 0.8800 | 0.8818 | 0.9258 | **0.3179** |
| Delicious | 0.7581 | 0.7592 | 0.7965 | **0.1782** |
| Imdb | - | 0.5280 | 0.5351 | **0.3439** |

Table 8: Average Precision ↑

| Dataset | GLFS | LRDG | FLFS | FALS |
|---|---|---|---|---|
| 20NG | 0.6913 | **0.6915** | 0.5661 | 0.6433 |
| Corel5k | 0.1693 | 0.1677 | 0.1292 | **0.2583** |
| HumanGO | 0.7878 | 0.7885 | 0.6258 | **0.7911** |
| Ohsumed | **0.6451** | 0.6430 | 0.5718 | 0.4055 |
| Tmc2007_500 | 0.7226 | 0.7198 | 0.6982 | **0.7654** |
| Yelp | 0.9195 | **0.9200** | 0.9037 | 0.8509 |
| Bibtex | 0.3577 | 0.3546 | 0.0763 | **0.3708** |
| Bookmarks | 0.1023 | 0.1016 | 0.0668 | **0.3197** |
| Delicious | 0.1709 | 0.1697 | 0.1254 | **0.2927** |
| Imdb | - | **0.5269** | 0.5178 | 0.3612 |

Table 9: Micro-AUC ↑

| Dataset | GLFS | LRDG | FLFS | FALS |
|---|---|---|---|---|
| 20NG | 0.7307 | 0.7287 | 0.6253 | **0.7699** |
| Corel5k | 0.5597 | 0.5569 | 0.5382 | **0.8374** |
| HumanGO | 0.8551 | 0.8553 | 0.7647 | **0.8705** |
| Ohsumed | 0.6231 | 0.6204 | 0.5317 | **0.6626** |
| Tmc2007_500 | 0.8262 | 0.8233 | 0.8099 | **0.9219** |
| Yelp | 0.8060 | 0.8106 | 0.7657 | **0.8442** |
| Bibtex | 0.6843 | 0.6828 | 0.5233 | **0.7039** |
| Bookmarks | 0.5609 | 0.5598 | 0.5354 | **0.6303** |
| Delicious | 0.6146 | 0.6140 | 0.5960 | **0.8171** |
| Imdb | - | 0.5540 | 0.5442 | **0.6301** |

