# OpenReview forum: "Feature Attribution for Label-Specific Feature Selection in Multi-Label Learning"
_ICLR.cc/2026/Conference — Submitted to ICLR 2026_

### Official Review · Reviewer_QrAt · 2025-10-15

**Soundness:** 3
**Presentation:** 3
**Contribution:** 2
**Rating:** 4
**Confidence:** 4

**Summary:**

This paper introduces FALS, a rapid method for label-specific feature selection in multi-label learning. The core idea is to leverage a lightweight, shallow neural network for feature attribution. It assesses feature importance by computing the gradient of each label's output with respect to each input feature. This approach efficiently identifies the optimal feature subset for each label in parallel, addressing the high computational cost of traditional label-specific methods.

**Strengths:**

The core methodology is clear, simple, and effective. Furthermore, the paper provides a theoretical advantage for the label-specific approach over joint feature selection from an information-theoretic perspective.

**Weaknesses:**

Please see Questions.

**Questions:**

1)The method is built upon feature attribution. However, the introduction does not provide a clear rationale for selecting this technique as the foundation. Could the authors elaborate on this choice?
2)FALS evaluates feature gradients independently of the final classifier. Does the proposed method account for or capture potential feature interactions?
3)The complexity analysis states that FALS requires only T=10 epochs, whereas other algorithms need 100-1000 iterations. Could the authors provide a more detailed justification for this significant difference? Why is the shallow network considered sufficiently trained for attribution in so few epochs?
4)The experiments include three single-label selection methods (e.g., RF) for comparison. What is the significance of including these baselines? Additionally, only two directly related label-specific methods were compared. Are there other recent or classic algorithms in this category that could be included for a more comprehensive comparison?
5)Regarding the box plots in Figure 2, Please clarify the meaning of the graphical elements. Does the red line represent the mean or median? What do the upper and lower bounds signify? Also, outliers (circles) are shown only for Hamming Loss. What do these circles represent, and why do they appear exclusively for this metric?
6)Regarding the third conclusion from the analysis of Figure 4: The paper claims that H=sqrt(DC) performs best. However, the plots for Ranking Loss and Average Precision suggest that performance is optimal when the number of layers is 1 (i.e., H=0) or when H=D or H=C, not necessarily H=sqrt(DC). Could the authors clarify this discrepancy?
7)The theoretical analysis in the appendix (proof of Theorem 2) relies on the assumption of label conditional independence. This assumption is often violated in multi-label learning where labels are correlated. To what extent does this violation affect the theoretical guarantees and practical performance of FALS?

---

### Official Review · Reviewer_9AqB · 2025-10-27

**Soundness:** 2
**Presentation:** 2
**Contribution:** 2
**Rating:** 4
**Confidence:** 3

**Summary:**

This paper introduces FALS (Feature Attribution based Label-specific Selection), a method for multi-label feature selection. The approach uses gradients from a simple two-layer perceptron (TLP) to score and select features for each label independently. The authors claim this method is highly efficient while achieving competitive performance.
This paper should be rejected because (1) the experimental validation lacks rigor, (2) the method's stability and sensitivity to its hyperparameters are not explored, and (3) Its contributions may lean toward incremental engineering heuristics rather than principle-based algorithmic breakthroughs.

**Strengths:**

1.Simplicity：The method's simplicity is a major advantage; relying on a shallow MLP makes it easy to understand and implement.
2.Computational Efficiency: The complexity analysis in Table 1 and the empirical runtime results in Table 4 show that FALS can be orders of magnitude faster, particularly on large datasets with many instances or labels.

**Weaknesses:**

1. Lack of Experimental Rigor and Insufficient Validation
1.1 Weak Protocol: The paper uses a single, fixed 2:1 stratified split for training and testing. This approach is highly susceptible to sampling bias and provides a much less reliable estimate of generalization performance than the standard five/ten-fold cross-validation used in related work (e.g., [1] [2]).
1.2 Incomplete Metrics: The evaluation omits key multi-label metrics like Coverage, Macro-F1, or Macro-AUC. These are crucial for understanding performance, especially on imbalanced datasets, which are common in multi-label learning.

2.Unexplored Stability and Incremental Contribution
2.1 The entire output of this method (feature ranking) relies solely on the gradients of a global optimal model (TLP) trained for only $T=10$. The selected features may merely be the result of insufficient training or overfitting in a specific model, rendering this method unreliable.
2.2 The "weak network" argument reads as a post-hoc justification for a simple model, not a well-motivated design choice. Figure 4 shows that performance actually decreased as the number of layers increased, a phenomenon that has not been adequately explained.

References：
[1] Zhu Y, Kwok J T, Zhou Z H. Multi-label learning with global and local label correlation[J]. IEEE Transactions on Knowledge and Data Engineering, 2017, 30(6): 1081-1094.
[2]Mao J, Wang W, Zhang M L. LIMIC: Label Specific Multi-Semantics Metric Learning for Multi-Label Classification: Global Consideration Helps[C]//IJCAI. 2023: 4055-4063.

**Questions:**

1.The entire method depends on meaningful gradients from the TLP. Can you provide a rigorous sensitivity analysis showing that the final feature rankings are stable across different random initializations, training epochs, and hyperparameters?
2.Given the model's simplicity, how would FALS perform on synthetic or real-world datasets specifically designed with highly non-linear decision boundaries or complex, higher-order feature interactions? The current experiments seem to avoid such challenging scenarios where the "weak learner" assumption would likely break down.
3.It is recommended to use k-fold cross-validation to update the experiment.
4.Key concepts are sometimes introduced without sufficient motivation. For example, the choice of hidden layer size $H=\sqrt{DC}$   is presented as an "empirical approach" without justification or reference. (line116)

Things to improve the paper that did not impact the score:
1.Tables 3 and 4 use numerical labels for dataset names. These datasets are familiar to researchers in the multi-label domain, and using numerical labels actually makes reading them less convenient. The significance test visualization (Figure 6) is very helpful. However, please consider adding text labels with the algorithm names to the nodes. This would make the plots much easier to interpret without needing to refer back to a separate list.
2.Label Independence Assumption in Theory: The theoretical justification connecting attribution to Fisher information relies on the assumption that labels are conditionally independent (a core aspect of the proof for Theorem 2). This contrasts sharply with principled multi-label methods, such as LIMIC [2], which explicitly model label co-occurrence to improve the learning process.

---

### Official Review · Reviewer_fN3q · 2025-10-29

**Soundness:** 3
**Presentation:** 3
**Contribution:** 1
**Rating:** 4
**Confidence:** 3

**Summary:**

This paper proposes a fast feature selection method for specific labels based on feature attributes. It employs feature attribution techniques from the interpretable domain to quantify each feature's contribution to prediction, achieved by utilizing the gradient of the output relative to the input. The experiments are comprehensive, and the model demonstrates favorable performance.

**Strengths:**

This paper features a well-structured and clearly organized framework, presenting a straightforward proof of the relationship between fisher information and multi-label feature attribution. The explanation is accessible and easy to understand. It also conducts a substantial number of comparative experiments, demonstrating thoroughness, and provides an intuitive analysis of time complexity.

**Weaknesses:**

In my view, this paper simply applies feature attribution methods from the interpretable domain directly. Moreover, it employs the most basic direct gradient attribution. The theoretical section merely calculates partial derivatives of the output with respect to each input feature while linking them to fisher information. Furthermore, simple direct gradient attribution has significant drawbacks. Nonlinear functions inevitably exhibit phases of gradient saturation, during which gradients tend toward zero, thereby violating sensitivity criteria. Direct gradient attribution focuses on the input-output relationship at a single point in the current input. When a feature's value happens to be in a gradient saturation phase, the attribution weight assigned to that feature is often negligible, even though the feature itself is not unimportant. Therefore, directly using gradient attribution presents significant problems.

**Questions:**

1. This paper employs an empirical average of individual instance gradients on the training set to obtain the final feature score matrix. Why is this approach taken? I believe weighting should be applied. Instances with more label 1s should receive lower weights, as these instances are more ambiguous. Conversely, instances with fewer label 1s should receive higher weights, as they better highlight certain labels.
2. Regarding feature attribution, this paper should compare attribution methods based on backpropagation, such as DeepLift and layer-wise elevance propagation (LRP), with integral gradient (IG) algorithms. I believe using IG is more reasonable than direct gradient attribution.

---

### Author Response · Authors · 2025-11-30
**Revised paper uploaded for review**

We have carefully addressed all reviewer comments and uploaded a fully revised manuscript. We added a new subsection, **4.3 In-Depth Analysis of Potential Risks in FALS**, along with additional explanations linked in the Appendix. The previous 4.3 has been merged into 4.2 or moved to Appendix, as those contents are less important than the new in-depth analysis.

The only remaining concern may be that FALS appears overly simple. However, our intention is to solve the multi-label feature selection problem using a simple yet effective approach. Existing algorithms continually introduce more and more penalty terms into their optimization objectives, which leads to slower training and a higher risk of overfitting. In contrast, **FALS achieves superior or comparable performance with only about 10% runtime**.

I apologize for the late response, but the experiments involving ten baselines under ten random runs (mentioned in Sec. 4.3), together with the 10-fold cross-validation studies, were extremely time-consuming.

---

> ### Author Response · Authors · 2025-11-30
> **Answer 1 - 5**
>
> We summarized the reviewers’ comments (where #1-W1 denotes Reviewer 1’s Weakness 1, #2-Q1 denotes Reviewer 2’s Question 1, and so on).
>
> 1.**Limited novelty**: The method simply uses a shallow neural network and its gradients to perform multi-label feature selection (#1-W1, #2-W2.2). Relatedly, why do the authors use such a simple model for multi-label feature selection? (#2-W2.2, #3-Q1).
>
> While we understand the reviewer’s concern that a shallow perceptron may appear simple from a deep-learning perspective, we emphasize that research communities often prioritize different objectives. In traditional machine learning for tabular data, optimization-based approaches remain highly competitive and widely adopted.
>
> In recent multi-label feature selection methods, it is common to incorporate increasingly complex regularization terms (e.g., label correlation, instance similarity, feature similarity). Although such designs enrich the model, they also introduce additional hyperparameters, increase computational cost, and potentially aggravate overfitting risks.
>
> In contrast, our work demonstrates that a very lightweight network combined with simple gradient-based attribution can outperform several sophisticated baselines while requiring only ~10% of their running time. We believe this finding—showing that simplicity can be effective and efficient—is itself of practical and methodological significance.
>
> 2.**Choice of baselines**: (1) Backpropagation-based attribution methods (such as IG, DeepLIFT and LRP) should be included (#1-Q2). (2) Why are single-label feature selection methods used, and (3) are there more recent label-specific methods that should be compared? (#3-Q4)
>
> (1) We added new experiments comparing FALS with backpropagation-based attribution methods, including IG, DeepLIFT and LRP. These results are reported in Section 4.3 (especially 4.3.2). The empirical evidence shows that FALS remains competitive or superior, confirming its effectiveness.
>
> (2) FALS produces a label-specific feature subset for each label. Structurally, this is similar to applying feature selection independently for each binary classification task. Therefore, we compare FALS with representative and advanced single-label feature selection methods to ensure fairness.
>
> (3) As far as we know, recent “label-specific” methods for multi-label learning tend to adopt embedded approaches and do not explicitly output per-label feature subsets. For this reason, direct comparison is difficult. Nevertheless, multi-label feature selection methods still employ the idea of label-specific, for example GLFS (paper titled “Group-preserving label-specific feature selection for multi-label learning”), have already been included in our experiments.
>
> 3.**Gradient saturation issue** (#1-W2).
>
> FALS employs a shallow neural network with very mild nonlinearity, which substantially reduces the likelihood of gradient saturation.
>
> Furthermore, several attribution methods mentioned by the reviewer (IG, DeepLIFT, LRP) inherently mitigate or completely avoid saturation effects. As shown in Figure 3, the empirical performance of our method remains superiority, indicating that gradient saturation does not significantly affect FALS in practice. Additional discussion and supporting analysis are provided in Section 4.3.2.
>
> 4.**The issue of imbalanced labels** (the positive labels are in minority) (#1-Q1). In other words, should instances with more positive labels be weighted highly when computing feature scores, rather than simply averaging over all instances? (#1-Q1)
>
> Although multi-label datasets often exhibit severe label imbalance, our empirical observations show that the positive and negative gradients produced by FALS are quite balanced. In other words, features that push the prediction toward the positive labels and those toward negative labels appear in comparable proportions. Consequently, we did not find it necessary to introduce additional label-balancing or instance-weighting strategies. The gradient and label-distribution visualizations are provided in Figure 5, and detailed analysis is given in Section 4.3.4.
>
> 5.**No k-fold cross-validation** (#2-W1.1, #2-Q3), and Missing imbalanced-learning metrics (#2-W1.2).
>
> Using k-fold cross-validation on multi-label datasets dramatically increases the computational cost. To avoid unintended variability caused by repeated random splits, we adopted a fixed stratified train-test split, which is the standard partition provided by the public repository KDIS. This ensures reproducibility and prevents selective reporting.
>
> We acknowledge that k-fold cross-validation offers more statistically robust estimates. Therefore, we added additional experiments comparing FALS with three multi-label feature-selection methods under several imbalance-aware metrics. Due to the high computational cost, we restricted the comparison to 3 latest multi-label feature selection. The results are reported in the revised manuscript Appendix D.4.

---

> > ### Author Response · Authors · 2025-11-30
> > **Answer 6 - 11**
> >
> > 6.**Potential instability of the method**: Should more baselines be used to verify the stability of the selected features? (#2-W2.1, #2-Q1)
> >
> > We appreciate the reviewer’s concern regarding stability. In the revision, we added a detailed analysis. Figure 4 reports the similarity between the feature subsets selected by FALS and those selected by other baselines, while Figure 3 shows the corresponding performance comparison. Together, these results indicate that FALS is reasonably stable and achieves superior predictive performance. Additional discussion is provided in Section 4.3.1.
> >
> > 7.**Possible failure in more challenging, highly non-linear scenarios** (#2-Q2).
> >
> > We agree that FALS is primarily designed for multi-label tabular data. According to the no-free-lunch theorem, no single model performs best across all problem settings. Our supplemental experiments in Section 4.3 demonstrate that, on multi-label tabular datasets, FALS outperforms IG, DeepLIFT and LRP. It remains an interesting and reasonable hypothesis that these more expressive approaches may outperform FALS on substantially more complex data, and we plan to explore this direction in future work.
> >
> > 8.**No theoretical justification for the hidden-layer width** (#2-Q4, #3-Q6).
> >
> > The hidden-layer width is chosen as the geometric mean of the input and output dimensions, following a common design principle that the number of neurons should not change too abruptly across layers. This choice is intuitive rather than theoretically derived. As shown in Appendix D.3.3, the performance of FALS is relatively insensitive to this hyperparameter, which suggests that the precise width is not critical. Given this robustness and the simplicity goal of FALS, we maintain this heuristic choice in the final version.
> >
> > 9.**Whether label correlations and feature interactions are considered** (#3-Q2, #3-Q7, #2-Q).
> >
> > In the revised manuscript, we designed an experiment to demonstrate that FALS indeed considers label correlation. FALS provides a multi-label model and then performs feature selection for each label. In contrast, the comparison method FALSb constructs multiple independent single-label binary classification models, each followed by feature selection using its corresponding binary perceptron. Clearly, FALSb completely ignores label correlations. The results in Section 4.3.3 show a significant difference between FALS and FALSb, indicating that FALS does account for label correlation to some extent. The underlying reason is that FALS builds a MIMO perceptron mapping features to labels, which integrates the interactions among different labels and features during training. Therefore, we believe that FALS takes into consideration both feature interactions and label correlations.
> >
> > 10.**Number of training epochs**: Why can FALS use only 10 epochs? (#3-Q3)
> >
> > Our experiments verified that 10 epochs are sufficient for FALS. Section 4.3.1 demonstrates that models with more epochs never help to enhance performance. For traditional optimization-based models, we used the default settings provided in their papers and source codes, where the number of training epochs is typically greater than 1000 as part of their stopping criteria.
> >
> > 11.**Other minor issues**, mainly related to figure formatting and descriptions (#2, #3-Q5).
> > We will further polish the camera-ready version to ensure the overall readability of the paper.

---

### Meta-Review · Area_Chair_XF3P · 2026-01-06

**Summary:**

This paper proposes FALS, a label-specific feature selection method for multi-label learning that employs a lightweight shallow neural network and gradient-based feature attribution to select label-specific feature subsets for each label. The main strengths of this paper lie in its concise structure and high computational efficiency, as it avoids complex regularization terms and achieves comparable performance to baseline methods with significantly reduced runtime. However, after a comprehensive evaluation of the paper, reviewer comments, and author rebuttals, it is determined that the paper fails to address core issues related to novelty, theoretical rigor, and experimental completeness-critical criteria for acceptance at this conference. The unresolved limitations undermine the paper’s scientific contribution and generalizability.

**Reviewer Concerns:**

**Addressed Concerns**

The authors provided partial responses to several reviewer inquiries:

1. **The rationale for using shallow networks:** The authors justified the simplicity of the model by highlighting that complex regularization terms in existing methods increase computational cost and overfitting risk, while their lightweight methods achieves efficiency without sacrificing performance. This addresses the core concerns regarding model simplicity raised by Reviewer fN3q and QrAt.

2. **Gradient saturation:** The authors noted that the shallow MLP structure mitigates saturation and referenced empirical results in Figure 3 demonstrating stable performance, partially addressing the concern of Reviewer fN3q.

3. **Inclusion of baseline methods:** The authors clarified that single-label feature selection methods are included due to structural similarities with label-specific selection, and recent label-specific methods were excluded as they lack explicit per-label feature subset outputs. They also added relevant methods like GLS to the experiments, partially addressing the baseline-related concerns of Reviewer QrAt.

4. **Label correlations:** The authors designed supplementary experiments (FALS and FALSb) to show that FALS implicitly accounts for label correlations through the MLP's multi-output mapping, partially addressing the concerns of Reviewer 9AqB and QrAt.

**Outstanding Concerns**

Critical issues remain unaddressed or inadequately resolved:

1. **Novelty and theoretical contribution:** All reviewers noted that FALS directly applies basic gradient attribution from the interpretable AI domain without substantial methodological innovation. The theoretical connection to Fisher information relies on the unrealistic assumption of label conditional independence (Reviewer 9AqB), which contradicts the nature of multi-label data where label correlations are prevalent. This undermines the theoretical guarantees of the method.

2. **Experimental rigor:** Key multi-label metrics (e.g., Coverage, Macro-F1, Macro-AUC) are missing (Reviewer 9AqB), limiting the assessment of performance on imbalanced datasets common in multi-label learning. Figure formatting and description issues remain unaddressed (Reviewer 9AqB and QrAt), hindering result interpretation.

3. **Method stability and hyperparameter sensitivity:** The authors' supplementary analysis of feature subset similarity is insufficient to fully address concerns about method stability (Reviewer 9AqB). Additionally, the choice of hidden-layer width (geometric mean of input and output dimensions) lacks theoretical justification and sensitivity analysis beyond empirical observation (Reviewer 9AqB and QrAt).

4. **Gradient attribution limitations:** The critical concern on gradient saturation raised by Reviewer fN3q—where irrelevant attribution weights may be assigned to important features in saturated regions—was not fully resolved. The authors' reference to mitigation by related methods (e.g., DeepLIFT, LRP) is irrelevant, as FALS uses direct gradient attribution without incorporating these enhancements.

**Reviewer Scores:**

* **Reviewer fN3q:** Original score (4: marginally below acceptance threshold). No substantial resolution of novelty and gradient attribution concerns would leave the score unchanged.

* **Reviewer 9AqB:** Original score (4: marginally below acceptance threshold). Unaddressed experimental rigor and theoretical assumption issues would leave the score unchanged.

* **Reviewer QrAt:** Original score (4: marginally below acceptance threshold). Inadequate justification for key design choices (epochs, hidden-layer width) and unresolved theoretical limitations would leave the score unchanged.

---

### Decision · Program_Chairs · 2026-01-26

Reject